

**Simulations of Organic Aerosol Concentrations during Springtime in the Guanzhong**
**Basin, China**
Tian Feng[1,2,3], Guohui Li[2,3*], Junji Cao[2,3*], Naifang Bei[1,2], Zhenxing Shen[1], Weijian Zhou[3], Suixin Liu[2,3], Ting
Zhang[2,3], Yichen Wang[2,3], Xuexi Tie[2,3], Luisa T. Molina[4]
[1]School of Human Settlements and Civil Engineering, Xi'an Jiaotong University, Xi'an, Shaanxi, China
[2]Key Lab of Aerosol Chemistry and Physics, Institute of Earth Environment, Chinese Academy of Sciences,
Xi'an, Shaanxi, China
[3]State Key Laboratory of Loess and Quaternary Geology, Institute of Earth Environment, Chinese Academy of
Sciences, Xi'an, China
[4]Molina Center for Energy and the Environment, La Jolla, CA, and Massachusetts Institute of Technology,
Cambridge, MA, USA
*Correspondence to: Guohui Li (ligh@ieecas.cn) and Junji Cao (jjcao@ieecas.cn)
**Abstract.** The organic aerosol (OA) concentration is simulated in the Guanzhong basin,
China from 23 to 25 April 2013 utilizing the WRF-CHEM model. Two approaches are used
to predict OA concentrations: (1) a traditional secondary organic aerosol (SOA) module; (2) a
non-traditional SOA module including the volatility basis-set modeling method in which
primary organic aerosols (POA) are assumed to be semi-volatile and photochemically
reactive. Generally, the spatial patterns and temporal variations of the calculated hourly
near-surface ozone and fine particle matters agree well with the observations in Xi'an and
surrounding areas. The model also yields reasonable distributions of daily $PM_{2.5}$ and
elemental carbon (EC) compared to the filter measurements at 29 sites in the basin. Filter
measured organic carbon (OC) and EC are used to evaluate OA, POA, and SOA using the
OC/EC ratio approach. Compared with the traditional SOA module, the non-traditional
module significantly improves SOA simulations and explains about 88% of the observed
SOA concentration. Oxidation and partitioning of POA treated as semi-volatile constitute the
most important pathway for the SOA formation, contributing more than 75% of the SOA
concentrations in the basin. Residential emissions are the dominant anthropogenic OA source,
constituting about 50% of OA concentrations in urban and rural areas and 30% in the
background area. The OA contribution from transportation emissions decreases from 25% in
urban areas to 20% in the background area, and the industry emission OA contribution is less
than 6%. The simulation results will facilitate the design of the air pollution control strategies
in the basin.
**Keywords:** SOA, $PM_{2.5}$, WRF-CHEM Model



## 1    Introduction

Atmospheric aerosols or fine particulate matters (PM$_{2.5}$) influence regional and global
climate directly by absorbing and scattering the solar radiation and indirectly by serving as
cloud condensation nuclei (CCN) and ice nuclei (IN) to modify cloud properties (Seinfeld
and Pandis, 2006). Elevated aerosols also exert adverse impacts on ecosystems and human
health, and reduce the visibility of the atmosphere to cause the haze formation, impairing air
quality (Cao et al., 2012a; 2012b; Greenwald et al., 2006; Seinfeld and Pandis, 2006).
OA constitutes one of the most important component of PM$_{2.5}$ in the atmosphere, with
the mass contribution to PM$_{2.5}$ ranging from 20% to 90% (Kanakidou et al., 2005; Zhang et
al., 2007). Traditionally, OA is categorized into primary and secondary OA on the basis of its
source and/or formation, referred to as POA and SOA, respectively. POA is emitted into the
atmosphere directly, while SOA forms through complicated chemical reactions of precursors
in the atmosphere. Volatile organic compounds (VOCs) emitted from anthropogenic or
biogenic sources undergo a series of oxidation and gas-particle partitioning to yield SOA,
which has been regarded as a traditional pathway of the SOA formation (Odum et al., 1996;
Pankow, 1994). Recently, semi-volatile POA has been identified to be oxidized continuously
to produce SOA in laboratory studies (Lipsky and Robinson, 2006; Shrivastava et al., 2006).
The mechanism has been parameterized into chemical transport models (Dzepina et al., 2009;
Lane et al., 2008; Li et al., 2011b; Murphy and Pandis, 2009; Robinson et al., 2007;
Shrivastava et al., 2008; Tsimpidi et al., 2010), significantly improving SOA simulations and
capable for closing the gap between the calculated and observed SOA concentrations.
China has been suffering severe air pollutions caused by rapid industrialization and
urbanization (Cao et al., 2007; 2005; 2012b; Guinot et al., 2007; He et al., 2015; 2001; Shen
et al., 2009; Tie et al., 2015; Yang et al., 2011; Zhang et al., 2015; 2013). Numerous studies
have shown that OA play an important role in the haze formation in China. Huang et al.



(2014) have reported that OA constitute a major fraction (30~50%) of the total $PM_{2.5}$ in
Beijing, Shanghai, Guangzhou, and Xi'an, and during the severe haze pollution event, SOA
contribute 30~77% of $PM_{2.5}$ and 44~71% of OA mass concentrations. Using the ACSM
(Aerosol Chemical Speciation Monitor) measurement analyzed by PMF (positive matrix
factorization), Sun et al. (2012) have showed that the oxygenated organic aerosols (OOA, a
surrogate of SOA) dominate OA composition, with a contribution of 64% on average from 26
June to 28 August 2011. Sun et al. (2013) have found that OA account for 52% of the total
non-refractory submicron particulate matters mass loading measured by ACMS during 2012
wintertime in Beijing. Cao et al. (2013) have reported that secondary organic carbon
constitutes 31% of the total carbon utilizing an EC tracer system and isotope mass balance
calculations during the MIRAGE-Shanghai (Megacities Impact on Regional and Global
Environment) campaign in 2009.

The Guanzhong basin (GZB) is located in northwestern China and nestled between the

Qinling Mountains on the south and the Loess Plateau on the north (Figure 1), with a
warm-humid climate. The rapid increasing industries and city expansions, as well as the
unique topography, have caused frequent occurrence of haze events in GZB (Shen et al., 2009;
2008). Measurements have shown that carbonaceous aerosols (OA and EC) constitute 48.6%
and 45.9% of $PM_{2.5}$ mass concentration in fall and winter, respectively in Xi'an, the largest
city of GZB (Cao et al., 2005). Abundant OA has been reported to be directly emitted into the
atmosphere from anthropogenic emissions, such as residential and transportation sources in
GZB (Cao et al., 2005). High SOA level has been observed in GZB during wintertime
(Huang et al., 2014). However, the source and formation of OA and especially SOA in GZB
still remain obscure. During the period from 20 to 26 April 2013, a field campaign was
conducted to identify the OA distribution and sources in GZB and an episode during 23-25
April was simulated to identify the OA sources in this study. Daily filter measurements at 29




sites in GZB were collected, and EC and OC were analyzed in $PM_{2.5}$, which provides a good
opportunity for better understanding OA and particularly SOA sources and formation in GZB.

The purpose of the present study is to investigate the formation and source

apportionments of OA and SOA in GZB during springtime using the WRF-CHEM model.
The WRF-CHEM model and its configuration are described in Section 2. The model results
and discussions are presented in Section 3, and the conclusions are summarized in Section 4.

**2    Model and Method**
**2.1    WRF-CHEM Model**

A specific version of the WRF-CHEM model (Grell et al., 2005) developed by Li et al.

(2011b; 2011a; 2012; 2010) at Molina Center for Energy and the Environment (MCE2) is
utilized to investigate the OA formation in GZB. This version employs a flexible gas-phase
chemical module and the CMAQ (version 4.6) aerosol module developed by US EPA
(Binkowski and Roselle, 2003). The dry deposition of chemical species is parameterized
according to Wesely (1989) and the wet deposition follows the method in CMAQ. The FTUV
module (Li et al., 2005; Tie et al., 2003) considering the impacts of aerosols and clouds on
photochemistry is used to calculate the photolysis rates. The ISORROPIA Version 1.7
(http://nenes.eas.gatech.edu/ISORROPIA/) is employed to the WRF-CHEM model to
simulate the inorganic aerosols.
**2.2    Secondary Organic Aerosol Modules**

Two kinds of SOA modules are utilized in the WRF-CHEM model simulations: a

traditional 2-product SOA module (T2-SOA module) and a non-traditional SOA module
(NT-SOA module) (Li et al., 2011b).

In the T2-SOA module, SOA concentrations are predicted from the oxidation of six

lumped organic species, including alkanes, alkenes, cresol, high-yield aromatics, low-yield





aromatics, and monoterpenes, following the method developed by Schell et al. (2001).
The NT-SOA module simulates SOA formation based on the volatility basis-set (VBS)
method (Donahue et al., 2006; Robinson et al., 2007). In the module, the POA is distributed
in logarithmically spaced volatility bins and assumed to be semi-volatile and photochemically
reactive (Li et al., 2011b). Nine surrogate species with saturation concentration ranging from
$10^{-2}$ to $10^{6}$ μg m$^{-3}$ at room temperature are selected to represent POA components following
(Shrivastava et al., 2008). The SOA formation from glyoxal and methlyglyoxal is
parameterized as a first-order irreversible uptake by aerosol particles with a reactive uptake
coefficient of $3.7\times10^{-3}$ (Volkamer et al., 2007; Zhao et al., 2006). Detailed information about
T2-SOA and NT-SOA modules can be found in Li et al. (2011b).
**2.3    Model Configuration**
In this study, a three-day episode from 23 to 25 April 2013 is simulated in association
with the filter measurements of PM$_{2.5}$, OC, and EC in GZB. The model is configured with a
horizontal grid spacing of 3 km and 200 × 200 grids which is centered at 34.25°N and 109°E
(Figure 1). In the vertical direction, we use thirty-five levels in a stretched vertical grid with
spacing ranging from 50 m near the surface to 500 m at 2.5 km above ground level and 1 km
above 14 km. The physics and dynamics of the configuration adopt the microphysics scheme
of Hong and Lim (2006), the Yonsei University planetary boundary layer scheme (Hong et al.,
2006), the land surface scheme of 5-layer thermal diffusion (Dudhia, 1996), the Dudhia
shortwave scheme (Dudhia, 1989) and the rapid radiative transfer model (RRTM) longwave
scheme (Mlawer et al., 1997). No cumulus parameterization is used due to the high horizontal
resolution. The NCEP 1°×1° reanalysis data are used for the meteorological initial and
boundary conditions. The chemical initial and boundary conditions are interpolated from
Model for OZone And Related chemical Tracers (MOZART) output with a 6-hour interval
(Horowitz et al., 2003). The spin-up time for the simulation is one day.



The anthropogenic emission inventory (EI) including agriculture, industry, power plant,
residential, and transportation sources is developed by Zhang et al. (2009). Figure 2 shows
the monthly POA and VOCs emissions in GZB along with the Xi'an urban area. Large
anthropogenic emissions are concentrated in Xi'an and surrounding areas. The POA from the
transportation source and biomass burning are redistributed following Tsimpidi et al. (2010).
The MEGAN model is used to on-line calculate the biogenic emissions in the model
(Guenther et al., 2006).
**2.4     Statistical Methods for Comparisons**
The mean bias (*MB*), root mean square error (*RMSE*), and index of agreement (*IOA*)
are used to evaluate the model performance in simulating gas-phase species and aerosols.
$$MB = \frac{1}{N}\sum_{i=1}^{N}(P_i - O_i) \tag{1}$$

$$RMSE = \left[\frac{1}{N}\sum_{i=1}^{N}(P_i - O_i)^2\right]^{\frac{1}{2}} \tag{2}$$

$$IOA = 1 - \frac{\sum_{i=1}^{N}(P_i - O_i)^2}{\sum_{i=1}^{N}\left(\left|P_i - \overline{O}\right| + \left|O_i - \overline{O}\right|\right)^2} \tag{3}$$

where $P_i$ and $O_i$ are the predicted and observed variables, respectively. N is the total
number of the predictions for comparison and $\overline{O}$ donate the average of the observation. The
*IOA* ranges from 0 to 1, with 1 showing a perfect agreement of the prediction with the
observation.

**3      Results and Discussions**
**3.1     Model Performance**
The meteorological fields are of essential importance for the simulation of chemical
species concentrations in time evolution and spatial distribution (Bei et al., 2008; 2010; 2012).



In the present study, the observations of temperature, pressure, relative humidity, and wind
components at 50 meteorological stations in GZB are assimilated in the WRF-CHEM model
simulations using the four-dimension data assimilation (FDDA) to improve the simulation of
meteorological fields. Model performance is validated using the hourly ozone ($O_3$) and $PM_{2.5}$
observations at 13 monitoring sites in Xi'an and surrounding areas, released by the Ministry
of Environmental Protection of China (China MEP), and daily filter measurement of $PM_{2.5}$,
EC, and OC at 29 sites in GZB.
**3.1.1 Hourly $O_3$ and $PM_{2.5}$ Simulations in Xi'an and Surrounding Areas**
Figures 3 and 4 provide the spatial patterns of observed and simulated near-surface $O_3$
and $PM_{2.5}$ mass concentrations at 08:00 and 15:00 Beijing Time (BJT) from April 23 to 25,
2013 in Xi'an and surrounding areas, along with simulated wind fields. The calculated $O_3$ and
$PM_{2.5}$ spatial distributions are generally consistent with the observations at the monitoring
sites. At 08:00 BJT, the near-surface winds are weak or calm, and the low planetary boundary
layer (PBL) also facilitates the accumulation of pollutants, causing observed and simulated
high near-surface $PM_{2.5}$ mass concentrations. The $PM_{2.5}$ mass concentration frequently
exceeds 150 μg m$^{-3}$, causing heavy air pollutions in Xi'an and surrounding areas. Weak solar
insolation slows the photochemical activities and the low PBL is also favorable for buildup of
emitted $NO_x$, significantly lowering the $O_3$ level at 08:00 BJT, and the calculated and
observed near-surface $O_3$ concentrations range from 20 to 30 μg m$^{-3}$. At 15:00 BJT, with the
development of PBL and enhancement of winds, the $PM_{2.5}$ mass concentrations are decreased
but still remain high level in Xi'an and surrounding areas on April 23 and 24. The strong
divergence at 15:00 BJT on April 25 efficiently disperses the $PM_{2.5}$ accumulated in the
morning and remarkably improves the air quality in Xi'an and surrounding areas. The $O_3$
mass concentration are substantially increased to more than 80 μg m$^{-3}$ at 15:00 BJT with the
enhancement of photochemical activities.



Figure 5 presents the temporal variations of calculated and measured $O_3$ and $PM_{2.5}$
concentrations averaged over 13 monitoring sites in Xi'an and surrounding areas from April
23 to 25, 2013. The WRF-CHEM model generally replicates the observed $O_3$ variations
during the episode, i.e., the occurrence of peak $O_3$ concentrations in the afternoon due to
active photochemical processes and rapid falloff during nighttime caused by the $NO_x$ titration.
The *MB* and *RMSE* are 7.1 μg m$^{-3}$ and 21.3 μg m$^{-3}$, respectively, and the *IOA* is 0.89. The
model considerably overestimates the observed $O_3$ concentration on April 23, perhaps due to
the high $O_3$ background transport. However, on April 25, the model notably underestimates
the peak $O_3$ concentration in the afternoon, which is caused by the simulated strong
divergence in Xi'an and surrounding areas (Figure 3). In Figure 5, the observed $PM_{2.5}$
variations are reasonably well reproduced by the model, although overestimations and
underestimations still exist. The *MB* and *RMSE* are 8.1 μg m$^{-3}$ and 23.9 μg m$^{-3}$, respectively,
and the *IOA* is 0.86. The observed and simulated $PM_{2.5}$ mass concentrations both show that
during the three-day episode, the air quality in Xi'an and surrounding areas gradually
improves, with the $PM_{2.5}$ concentration decreased from about 160 μg m$^{-3}$ in the morning on
April 23 to about 50 μg m$^{-3}$ in the afternoon on April 25. The $PM_{2.5}$ mass concentrations are
generally elevated in the morning during the episode, caused by weak or calm horizontal
winds and slow development of PBL.
**3.1.2 Daily $PM_{2.5}$ and EC Simulations in GZB**
Daily filter measurements of $PM_{2.5}$ and EC mass concentrations at 29 sites in GZB
(squares in Figure 1) are used to further verify the WRF-CHEM model simulations. Figure 6
shows the scatter plots of calculated and measured daily $PM_{2.5}$ and EC mass concentrations at
29 sites during the episode. The simulated daily $PM_{2.5}$ mass concentrations are generally in
agreement with the filter measurement over 29 sites. The $PM_{2.5}$ concentration averaged over
29 sites during the episode is 79 μg m$^{-3}$, close to the observed 87 μg m$^{-3}$. The *MB* and *RMSE*



of PM$_{2.5}$ mass concentrations are -7.4 μg m$^{-3}$ and 10.3 μg m$^{-3}$, showing a reasonable PM$_{2.5}$
simulation in GZB. The WRF-CHEM model slightly overestimates the observed EC mass
concentrations, with the *MB* of 0.2 μg m$^{-3}$ and the *RMSE* of 0.6 μg m$^{-3}$.
The calculated daily spatial patterns of PM$_{2.5}$ and EC mass concentrations are displayed
in Figure 7 along with the measurements at the 29 sites. The simulated distributions of PM$_{2.5}$
and EC mass concentrations are consistent well with the filter measurements. On April 23,
the eastern part of GZB is the most polluted area, with the daily PM$_{2.5}$ mass concentrations
exceeding 115 μg m$^{-3}$. The daily PM$_{2.5}$ mass concentrations are still high on April 24,
exceeding 75 μg m$^{-3}$ over most of the area in GZB. The air quality in GZB is considerably
improved on April 25, and the daily PM$_{2.5}$ mass concentrations are generally less than 75 μg
m$^{-3}$ in the north part of GZB. The EC distributions do not exhibit remarkable variations from
April 23 to 25, indicating that the EC levels are primarily determined by direct emissions. In
addition, although the daily PM$_{2.5}$ mass concentrations decrease substantially from April 23
to 25, the variations of EC mass concentrations are not so significant.
The simulated distributions of the column-integrated aerosol optical depth (AOD) at
550 nm are verified using the available measurements from the MODIS (Moderate
Resolution Imaging Spectroradiometer) aerosol level-2 product in Figure 8. The retrieved
AOD shows that GZB suffers severe air pollutions at 12:00 BJT on 24 April, and the most
contaminated area is Xi'an and surrounding areas with AOD more than 1.2. The model
successfully reproduces the retrieved AOD spatial pattern in GZB, but underestimates the
retrieved AOD in the north of GZB, which might be caused by the underestimation of dust
aerosols. At 11:00 BJT on 25 April, the model underestimates the retrieved AOD in the west
part of GZB, perhaps due to the biases of simulated relative humidity.
The WRF-CHEM model generally well reproduces O$_3$ and PM$_{2.5}$ concentrations in
Xi'an and surrounding areas, and the daily EC and PM$_{2.5}$ mass concentrations in GZB,





indicating that meteorological fields are well simulated and the emissions used are also
reasonable in the study.

### 3.2    Organic Aerosols in GZB

The OC/EC ratio approach is employed to evaluate the OA concentration from the filter
measured OC and EC concentrations (Strader, 1999). Previous studies using the OC/EC ratio
approach have been extensively conducted in China (Cao et al., 2003; 2004). Cao et al. (2007)
have analyzed the OC and EC concentrations in 14 cities over China in 2003 and proposed
primary OC/EC ratios of 2.81 for northern cities in winter, 2.10 for southern cities in winter,
1.99 for northern cities in summer, and 1.29 for southern cities in summer. To estimate the
primary OC (POC) and secondary OC (SOC) concentrations during April 2013 in GZB, a
POC/EC ratio of 2.4 is used to derive POC and SOC from OC and EC measurements.
Numerous studies have been performed to investigate the POA/POC and SOA/SOC ratios
(Aiken et al., 2008; Yu, 2011; Yu et al., 2009), which can be used to obtain OA
concentrations from measured OC concentrations. In the present study, a POA/POC ratio of
1.2 and a SOA/SOC ratio of 1.6 are adopted to obtain POA and SOA concentrations, as well
as OA concentrations.

### 3.2.1 OA Simulations from the T2-SOA and NT-SOA modules

Figure 9a shows that the scatterplot of the observed and simulated OA concentrations at
29 sites during the episode. Apparently, both the T2-SOA and NT-SOA modules reasonably
replicate the observed OA mass concentrations in GZB, and the OA difference between the
two modules is not remarkable, as shown in Figure 9a. The T2-SOA module overestimates
the observed OA concentrations by about 33.6% averaged over the 29 sites, and the NT-SOA
modules underestimates the observation by about 4.3%. Although the T2-SOA and NT-SOA
modules both reproduce comparable OA levels against the measurements, the simulated OA
components, i.e. POA and SOA, differ greatly between the two modules. Figures 9b and 9c



provide comparisons of the simulated POA and SOA concentrations from the T2-SOA and
NT-SOA modules with the measurements, respectively. The T2-SOA module overestimates
the measured POA by around 132.0%, and only explains about 9.4% of the observed SOA
concentration. The SOA simulations are comparable to Li et al. (2011b), in which the
T2-SOA module fails to yield sufficient SOA concentrations to match the observations by a
factor of 7. As a comparison, the NT-SOA module overestimates the observed POA by 17.5%
and explains around 87.7% of the observed SOA concentration, significantly improving the
POA and SOA simulations.
Figure 10 presents the temporal variations of simulated SOA mass concentrations in
Xi'an and surrounding areas from the T2-SOA and NT-SOA modules, respectively. The
diurnal variations from the two models agree well with each other, with peak occurrence at
noontime, caused by the enhanced photochemical activities. The NT-SOA module
remarkably improves the SOA yields during the entire episode to around 10 μg m$^{-3}$, about
tenfold increase compared with the simulations from the T2-SOA module.
**3.2.2 Urban, Rural and Background POA and SOA**
Since the NT-SOA module significantly improves the model performance in simulating
POA and SOA in GZB, Figure 11 displays the spatial distributions of OA, POA, and SOA
simulated by the NT-SOA module against the measurement in GZB. The simulated OA, POA,
and SOA patterns are generally in agreement with the observations, but the model frequently
underestimates the observations in the north part of GZB (Figure 11). Both the simulation
and the measurement show that the entire GZB is OA contaminated during the simulation
episode (20 μg m$^{-3}$ and above) (Figures 11a-c). POA is primarily concentrated in the central
part of GZB, directly linked to the source region (Figures 11d-f). However, SOA is dispersed
efficiently in the horizontal direction, showing the rapid aging process of OA. The
simulations reveal a progressive increase of SOA concentrations in background areas, which



is consistent with the measurements at the background site (red squares in Figures 11g-i).

The comparisons of the calculated and measured SOA mass fractions in OA at urban,

rural and background sites are displayed in Figure 12. The 29 sites are categorized into three
types based on their locations: 18 urban sites, 10 rural sites, and 1 background site, as shown
in Figure 1b. The average mass fractions over each type of the sites are colored red. The mass
fractions of SOA in OA at urban and rural sites are very close (around 44%~50% from the
observations and around 40% from the simulation on average), suggesting a similar OA aging
process in urban and rural areas of GZB. The SOA mass fraction at the background site is
much higher, which well agrees with the observations (around 85% from the observations
and around 70% from the simulation on average), indicating that the OA in background areas
undergoes long-time aging processes.

The SOA formation pathways considered in the NT-SOA module include (1) oxidation

of anthropogenic VOCs (ASOA), (2) oxidation of biogenic VOCs (BSOA), (3) oxidation and
partitioning of POA treated as semi-volatile (PSOA), and (4) irreversible uptake of glyoxal
and methylglyoxal on aerosol surfaces (GSOA). The SOA mass fractions in OA increase
from 35.1% at urban sites to 40.4% at rural sites, and sharply to 71.0% at the background site
(Table 1). PSOA dominate SOA mass concentrations at all sites, and its contribution to SOA
increases from 79.4% at urban sites to 88.8% at the background site, showing the continuous
aging process of OA (Figure 13). The SOA contribution from ASOA and GSOA decreases
from urban sites to the background site, showing the abatement of direct anthropogenic
impacts. At urban sites, ASOA, BSOA, and GSOA contribute comparably to the SOA mass
concentration. At the background site, the SOA contribution from ASOA and GSOA is very
low, less than 3%. The GSOA constitutes about 10% of SOA mass in the afternoon at urban
sites, close to the results of Li et al. (2011b) in Mexico City.



### 3.2.4 Contributions of Anthropogenic Emissions to OA


Sensitivity studies are conducted to verify the contributions from anthropogenic
emissions including industry, residential and transportation sources to the OA mass
concentrations during the episode. The factor separation approach (FSA) proposed by Stein
and Alpert (1993) is utilized to decompose the contribution from an individual source. The
simulation with all anthropogenic emissions is taken as the base case (referred to as BAS case)
to compare with the sensitivity studies. Four sensitivity studies are performed, including (1)
ANT case without all anthropogenic emissions, (2) RES case without the residential emission,
(3) IND case without the industry emission, and (4) TRA case without the transportation
emission in simulations. According to the FSA approach, the OA contribution from an
individual source, i.e., residential emissions, can be calculated as OA(BAS) – OA(RES).
Anthropogenic emissions dominate the OA level at the urban and rural sites, with the
OA contribution of 82.4% (18.2 $\mu$g m$^{-3}$) and 77.3% (12.8 $\mu$g m$^{-3}$), respectively (Table 2). At
the background site, the OA contribution from anthropogenic emissions is close to 60% (4.7
$\mu$g m$^{-3}$), showing that the background area is still substantially influenced by human activities
despite the far distance from the urban area. The residential emission is the most important
anthropogenic OA source at the urban and rural sites, with the OA contribution close to 50%,
indicating that reducing residential emission is an efficient approach for OA mitigation in
GZB. The OA contribution from the transportation source is 25.0% at the urban sites,
exceeding 20.3% at the rural sites and 19.8% at the background sites. The industry emission
is not an important OA source in GZB, with the contribution less than 6.1%. It is worth to
note that, the OA contributions from residential, transportation, and industry emissions are
comparable at the urban and rural sites, indicating that the OA source difference between the
rural and urban area has been rapidly diminishing due to urbanization and industrialization in
GZB.




## 4     Summary and Conclusions

A 3-day episode from 23 to 25 April 2013 is simulated in the Guanzhong basin, China using the WRF-CHEM model to investigate the SOA formation and verify the OA source. We use two SOA approaches to simulate OA: a traditional 2-product SOA module and a non-traditional SOA module including VBS modeling method and the SOA contribution from dicarbonyl compounds. Meteorological observations during the simulation period are assimilated using the FDDA method in WRF-CHEM simulations. Model results are compared with hourly $O_3$ and $PM_{2.5}$ measurements in Xi'an and GZB.

The WRF-CHEM model generally well simulates the spatial distributions and temporal variations of near-surface $O_3$ and $PM_{2.5}$, but biases still exist due to the uncertainties of meteorological fields and emission inventories. The model performs reasonably well in reproducing the distribution of the filter measured daily $PM_{2.5}$ and EC in GZB, but underestimates the observed $PM_{2.5}$ mass concentration by 7.4 μg m$^{-3}$ on average.

The OC/EC ratio approach is used to evaluate OA, POA, and SOA concentrations from the filter measured OC and EC concentrations. The traditional and non-traditional SOA modules both yields reasonable OA simulations compared with measurements, but perform differently in simulating POA and SOA. The traditional module overestimates the measured POA concentration by around 132.0% but underestimates the observed SOA concentration by a factor of 10. The non-traditional module overestimates the observed POA concentration by 17.5% and explains around 87.7% of the observed SOA concentration, significantly improving the POA and SOA simulations. Although the model can produce similar $PM_{2.5}$ simulations when using the traditional and non-traditional SOA modules, the results might cause misleading when the traditional SOA module is used to devise the $PM_{2.5}$ mitigation strategy.



Simulations from the non-traditional SOA module show that oxidation and partitioning

of POA which is treated as semi-volatile in the model dominate the SOA concentration in

GZB, with the SOA contribution exceeding 75% and also gradually increasing from urban

sites to the background site. The oxidation of anthropogenic and biogenic VOCs and the

irreversible uptake of dicarbonyl compounds do not constitute an important SOA formation

pathway in GZB, with the SOA contributions less than 10% generally. Anthropogenic

emissions are the dominant OA source at urban and rural sites, contributing over 70% of OA

concentrations. Residential emissions are the most important anthropogenic OA sources,

constituting about 50% of OA concentrations at urban and rural sites and 30.2% at the

background site. Transportation emissions make up 25.0% of the OA concentrations at urban

sites and decreasing to 19.8% at the background site. The OA contribution from industry

emissions is not significant, less than 6.1% in GZB.

Although the WRF-CHEM model reasonably well predicts the patterns and variations

of observed $O_3$, $PM_{2.5}$, and aerosol components, biases still exist. Considering uncertainties

from measurements, emissions, meteorological fields, and the SOA modules, future studies

need to be performed to further improve SOA simulations and OA source apportionment, to

provide the underlying basis for better understanding the haze formation and support the

design and implementation of emission control strategies in the Guanzhong basin.


**Data availability**: The real-time $O_3$ and $PM_{2.5}$ are accessible for the public on the website

http://106.37.208.233:20035/. One can also access the historic profile of observed ambient

pollutants through visiting http://www.aqistudy.cn/.





*Acknowledgements.* This work was supported by the National Natural Science Foundation of
China (No. 41275153) and supported by the "Strategic Priority Research Program" of the
Chinese Academy of Sciences,Grant No. XDB05060500. Guohui Li is also supported by the
"Hundred Talents Program" of the Chinese Academy of Sciences. Naifang Bei is supported
by the National Natural Science Foundation of China (No. 41275101).



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





Table 1 Simulated mass fractions of SOA in the total OA and the SOA contribution from
various formation pathways averaged over the simulation period at urban, rural and
background sites

| Duration | Site Type | SOA Mass Fraction in OA (%) | Mass Fraction in SOA (%) | | | |
|---|---|---|---|---|---|---|
| | | | PSOA[a] | ASOA[b] | BSOA[c] | GSOA[d] |
| April 23 | Urban | 32.5 | 77.9 | 5.8 | 5.6 | 10.6 |
| | Rural | 36.3 | 80.3 | 5.2 | 5.1 | 9.4 |
| | Background | 58.6 | 86.9 | 4.0 | 5.4 | 3.7 |
| April 24 | Urban | 34.8 | 80.5 | 6.0 | 8.5 | 5.0 |
| | Rural | 39.9 | 84.4 | 4.0 | 8.5 | 3.1 |
| | Background | 74.4 | 89.4 | 3.2 | 5.8 | 1.6 |
| April 25 | Urban | 37.3 | 79.5 | 6.2 | 9.9 | 4.4 |
| | Rural | 44.1 | 80.7 | 5.3 | 10.3 | 3.7 |
| | Background | 75.7 | 89.2 | 2.4 | 6.8 | 1.6 |

[a] SOA from oxidation and partitioning of POA treated as semi-volatile;
[b] SOA from oxidation of anthropogenic VOCs;
[c] SOA from oxidation of biogenic VOCs;
[d] SOA from irreversible uptake of glyoxal and methylglyoxal on aerosol surfaces.





Table 2 OA mass concentrations and contributions from anthropogenic emissions averaged
over the simulation period at urban, rural, and background sites

| | OA Mass ($\mu g\ m^{-3}$) | OA Mass Contribution (%) |
|---|---|---|
| **Urban** | | |
| Ant.[a] | 18.2 | 82.4 |
| Res.[b] | 10.4 | 47.2 |
| Tra.[c] | 5.5 | 25.0 |
| Ind.[d] | 1.3 | 6.1 |
| **Rural** | | |
| Ant.[a] | 12.8 | 77.3 |
| Res.[b] | 7.9 | 47.8 |
| Tra.[c] | 3.4 | 20.3 |
| Ind.[d] | 0.4 | 2.6 |
| **Background** | | |
| Ant.[a] | 4.7 | 58.6 |
| Res.[b] | 2.4 | 30.2 |
| Tra.[c] | 1.6 | 19.8 |
| Ind.[d] | 0.0 | 0.0 |


[a] Ant. stands for all anthropogenic emissions;
[b] Res. stands for residential emissions;
[c] Tra. stands for transportation emissions;
[d] Ind. stands for industrial emissions.





**Figure Captions**



Figure 1 Map showing (a) the location of Xi'an in China, (b) WRF-CHEM model
simulation domain with topography and (c) geographic distributions of 13 ambient
air quality monitoring stations (black triangles) and 29 enhanced sites with filter
measurements (squares). The filled red, blue, and green squares represent the urban,
rural, and background sites, respectively. The area surrounded by the white line in
(c) is defined as Xi'an and surrounding areas.

Figure 2 Geographic distributions of anthropogenic emissions of (a) primary organic aerosol
and (b) volatile organic compounds in the simulation domain. The black lines
present provincial boundaries in China.

Figure 3 Spatial patterns of calculated (contours) and observed (squares) $O_3$ concentrations
at 08:00 BJT and 15:00 BJT from 23 to 25 April 2013 along with wind fields (back
arrows).

Figure 4 Spatial patterns of calculated (contours) and observed (squares) $PM_{2.5}$
concentrations at 08:00 BJT and 15:00 BJT from 23 to 25 April 2013 along with
wind fields (back arrows).

Figure 5 Temporal variations of simulated (blue line) and observed (black dots) (a) $O_3$ and
(b) $PM_{2.5}$ concentrations averaged over 13 monitoring sites in Xi'an and
surrounding areas from 23 to 25 April 2013.

Figure 6 Comparisons between the predicted and measured daily (a) $PM_{2.5}$ and (b)
elemental carbon mass concentrations at 29 sites from 23 to 25 April 2013. The 1:1,
1:2 and 2:1 lines are plotted as dotted lines.

Figure 7 Spatial distributions of calculated (contours) and observed (squares) daily $PM_{2.5}$
(left column) and EC (right column) concentrations from 23 to 25 April 2013.

Figure 8 Spatial patterns of simulated (contours) and retrieved (squares) aerosol optical
depth at 550 nm from MODIS satellite (a) at 12:00 BJT on 24 April and (b) 11:00
BJT on 25 April 2013

Figure 9 Scatter plots of the simulated (a) OA, (b) POA, and (c) SOA from the traditional
(blue spots) and non-traditional (red spots) SOA modules against the observations
at 29 sites from 23 to 25 April 2013.

Figure 10 Temporal variations of the simulated SOA concentrations from the traditional (blue
spots) and non-traditional (red spots) SOA modules in Xi'an and surrounding areas
from 23 to 25 April 2013.

Figure 11 Spatial distributions of calculated (contours) and observed (squares) daily OA (left
column), POA (middle column), and SOA (right column) mass concentrations from
23 to 25 April 2013. Red squares in (g), (h), and (i) show the location of the
background site.




Figure 12 Comparisons between the predicted and measured daily SOA mass fraction in OA
at urban, rural and background sites during the simulation episode
Figure 13 The contributions of different formation pathways to SOA levels averaged over the
simulation episode at (a) urban, (b) rural and (c) background sites.





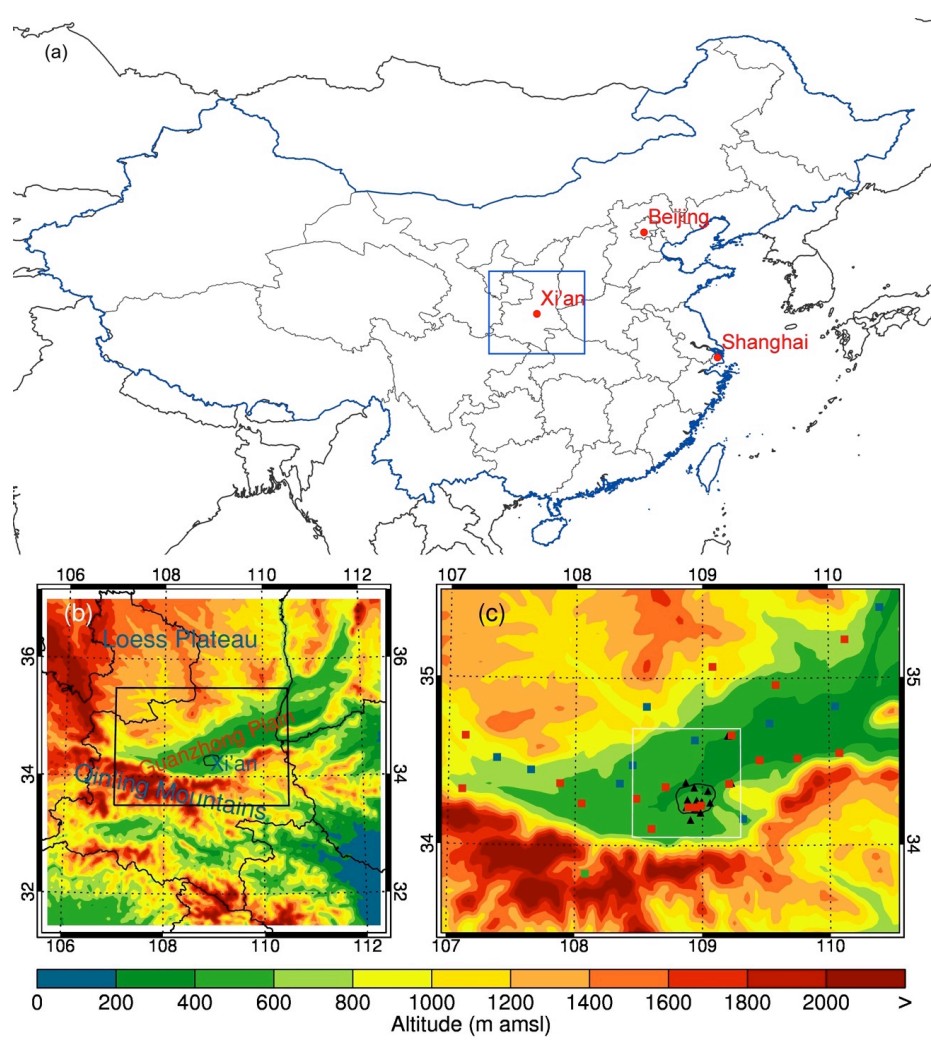

Figure 1

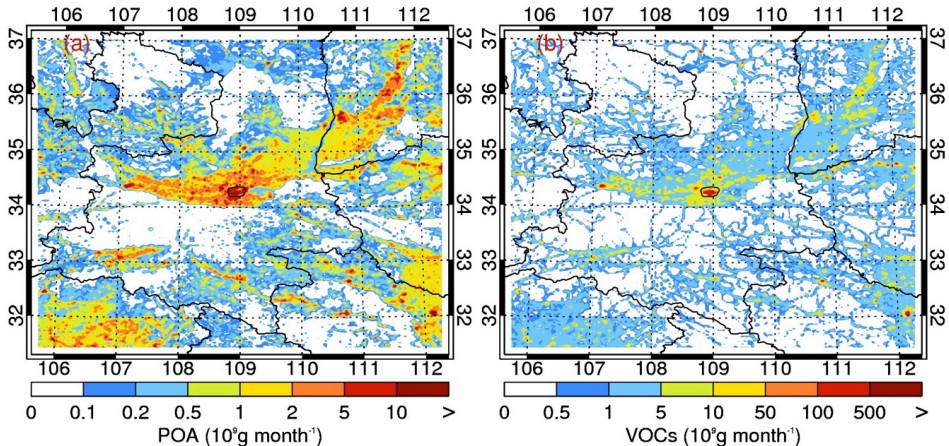

Figure 2





Figure 3





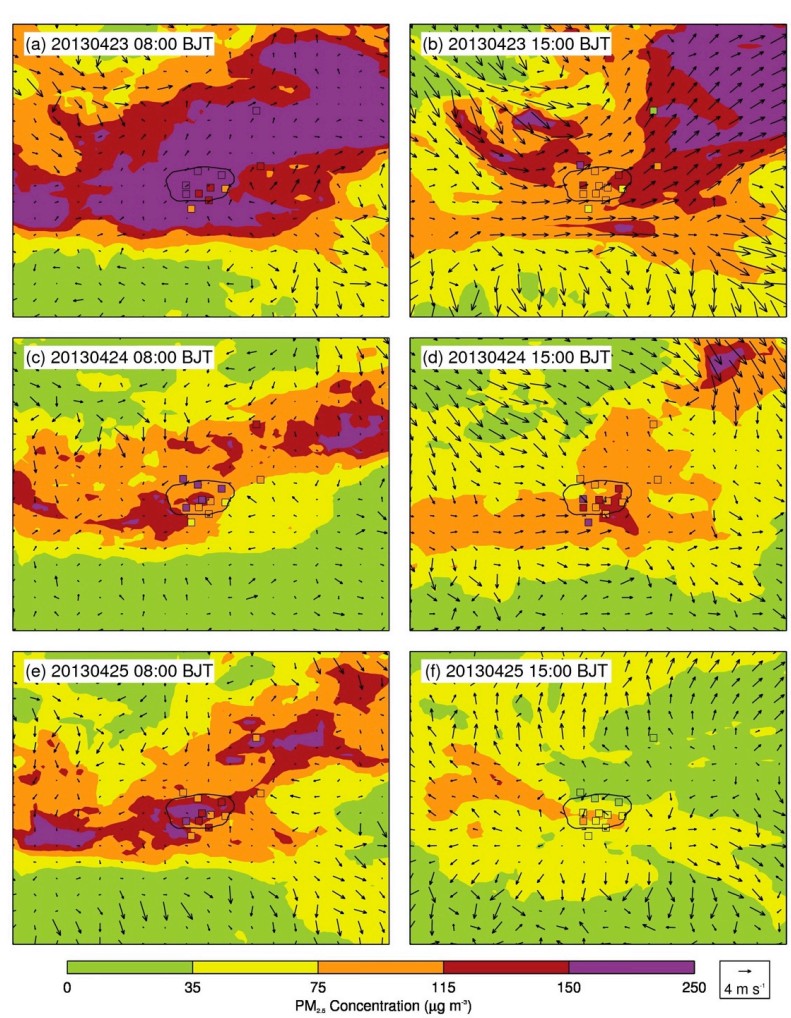

Figure 4



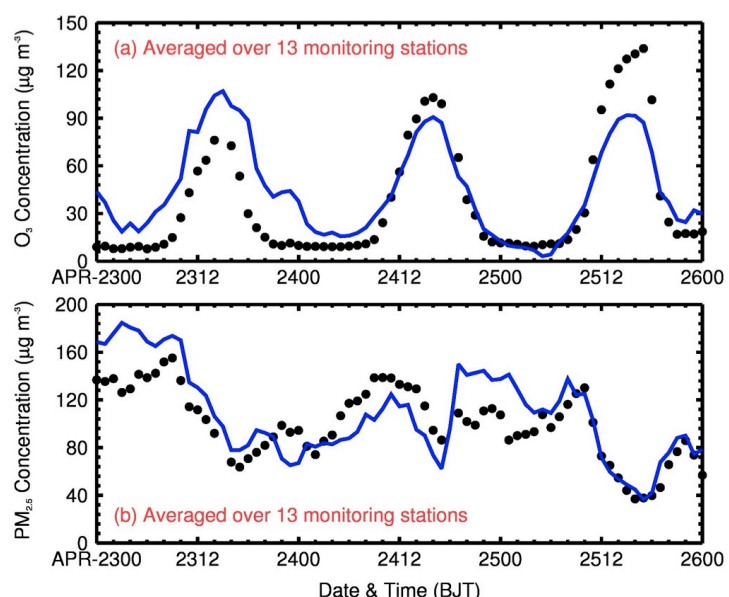

Figure 5





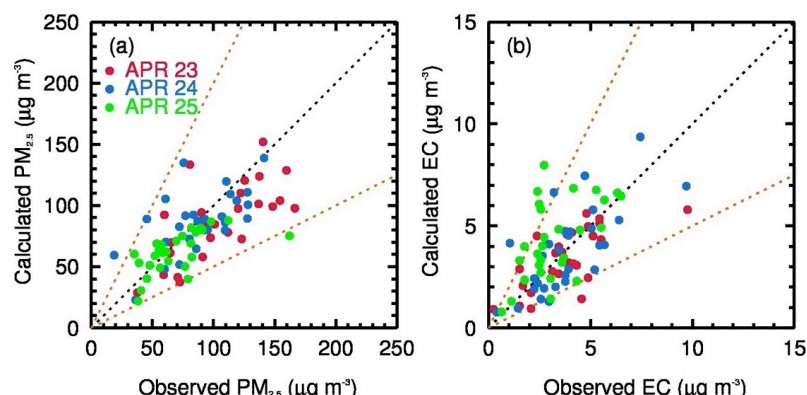

Figure 6





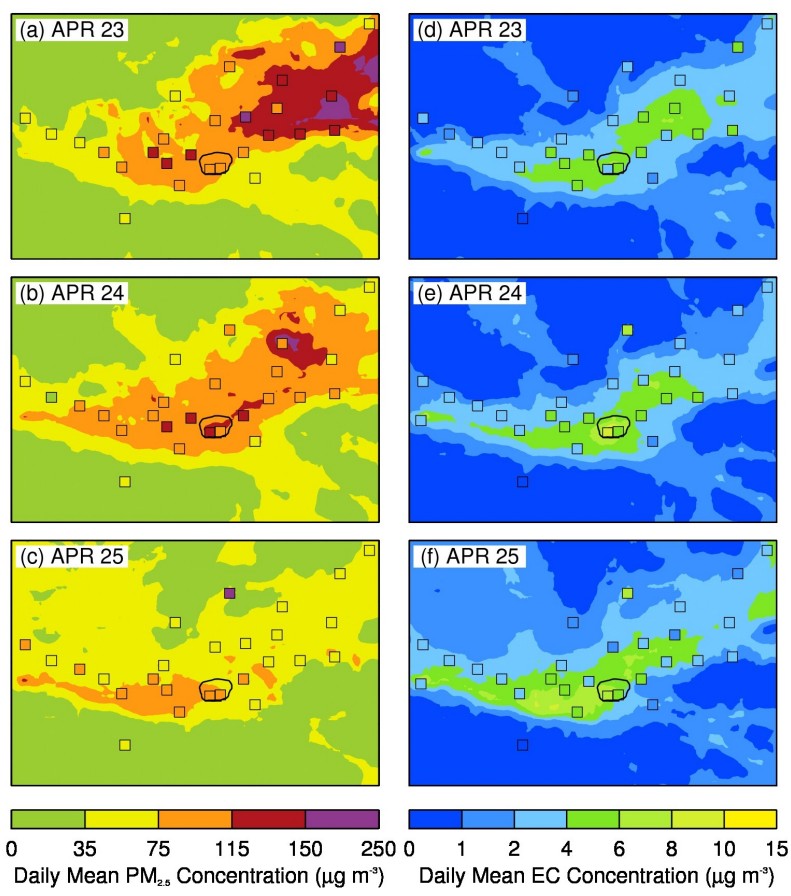

Figure 7




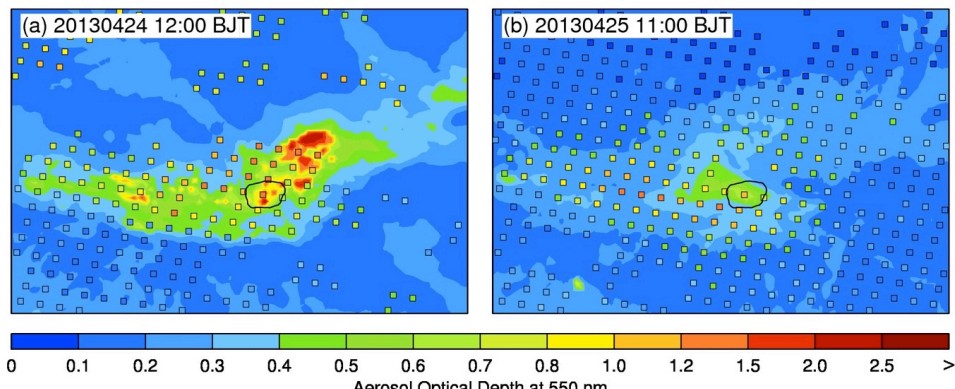

Figure 8





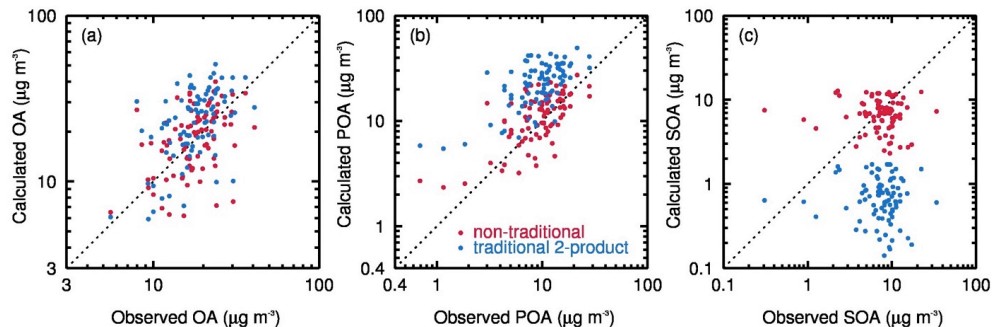

Figure 9





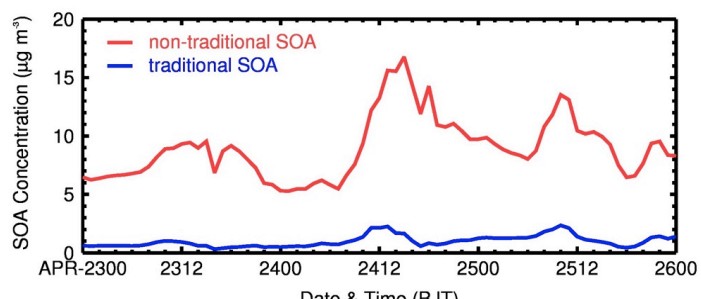

Figure 10






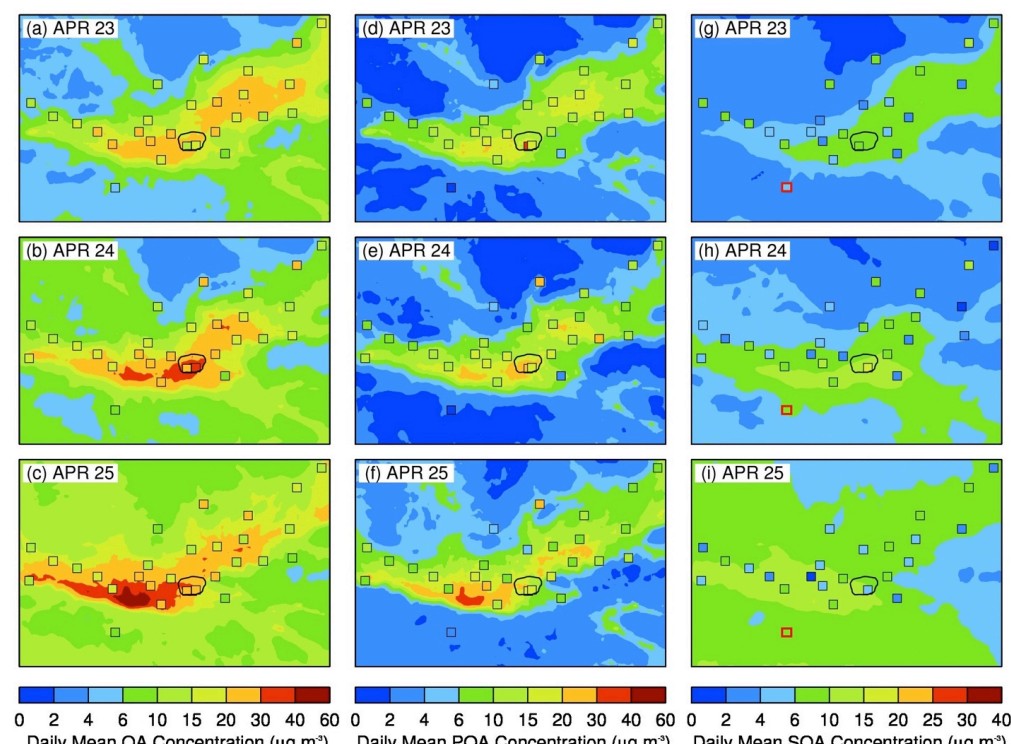

Figure 11






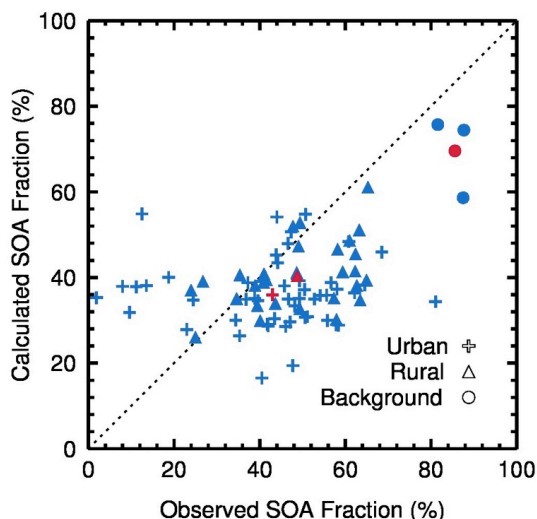

Figure 12





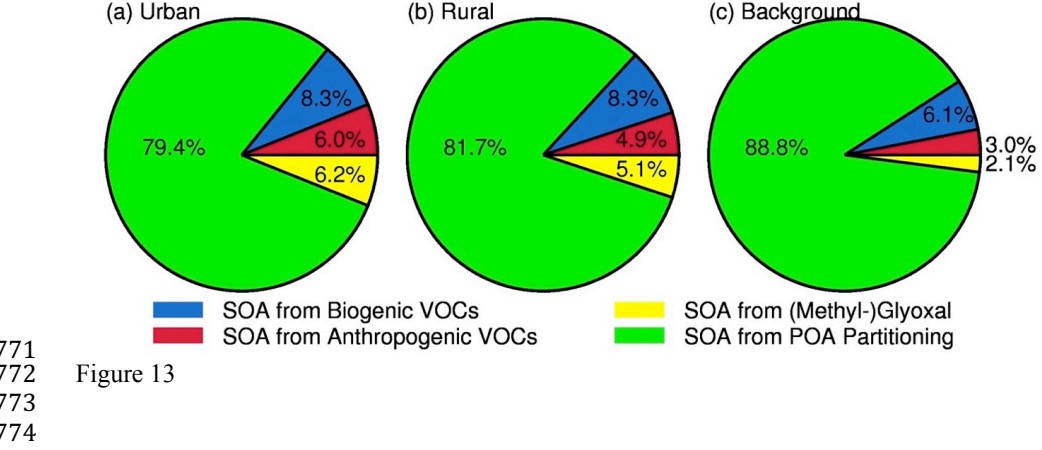

Figure 13