# Peer review of "Simulations of Organic Aerosol Concentrations during Springtime in the Guanzhong 1 2 Basin, China"

_Atmospheric Chemistry and Physics, 2016_

## Referee Comment (RC1) · Anonymous Referee #2 · 3 May 2016

This manuscript is clearly written and on an important topic on SOA simulation. It is easy to follow and some of its conclusions are interesting. But I am not happy respecting to following points. 1) The model and the method used in this application are nearly the same as in Li et al. (2011b), and one major conclusion is obvious that NT-SOA produces higher SOA than T2-SOA, because the total amount of material (POA+ SVOC+ IVOC) introduced in the NT-SOA module is 7.5 times the particle-phase POA emissions. Besides, many studies have already shown that VBS produces higher SOA than the traditional 2-product SOA module. 2) It does not clearly show whether diurnal (time) variation in SOA concentration is improved by including VBS than the traditional 2-product SOA module. NT-SOA module contains more precursors and production processes, is it good or not if "The diurnal variations from the two models agree well with each other, with peak occurrence during noontime, caused by the enhanced photo-

chemical activities." as shown in Figure 10? 3) The model performances are evaluated against surface observed O3, PM2.5, EC and OA, but the importance is not clearly seen to compare with satellite derived AOD. Contribution of OA or SOA to AOD is not significant, and it is not clearly stated in the manuscript how the model calculates AOD, and AOD values depend on aerosol number and size distribution, mixing state and air humidity, which are beyond the scope of this study. So good agreement between simulated and satellite retrieved AOD does not imply the model simulates PM2.5 and OA well.

---

## Referee Comment (RC2) · Anonymous Referee #3 · 19 May 2016

**General comments**

The paper is well structured and the chosen figures illustrate the results well. The description of the methods used is clear. Previous work is clearly described. The main limitation is that the present study presents simulations of 3 days lenght, which does not allow for drawing general conclusions, e.g. for a whole season. I suggest that the authors clearly describe this limitation and make it clear throughout the manuscript, and justify why and how the results are still important. Furthermore, I would suggest to include a small section on how the model performs in simulating meteorological variables that are important for simulating aerosols. This would help be more consistent on conclusions regarding the simulated meteorology. I suggest to revise the conclusions especially with regards to these two points, aiming at consistency with the results

stated in the paper. Finally, I suggest to carefully check the language especially in the introduction of the paper. For further specific comments please see below.

**Specific comments**

- **Page 2, line 49:** remove complicated

- **Page 3:** the description of the diffferent contributions of OA to PM2.5 is hard to follow, formulate it more understandably; a table summarizing previous results might help

- **Page 3, line 86:** why does the simulation not cover the whole period of the measurement campaign?

- **Page 4, line 89:** OA sources and SOA formation

- **Page 4, line 91:** Specify the research question, investigate is very broad. Also, OA and SOA is not investigated during springtime, but for three days in spring-time. Can you deduct general conclusions from a three day simulation for spring-time? If so, why? Is the simulated period particularly typical for springtime conditions?

- **Page 4, line 102 and following:** You simulate wet deposition, but you do not present any results on how well precipitation is simulated by the model; or if there even is precipitation. This should be included.

- **Page 5, line 134:** The jump from the boundary condition resolution to the model resolution is big, and I understand that you do not use a nested setup. How far is the studied area away from the boundaries of the domain? Please specify this in the text.

- **Page 6, line 140:** What is the temporal resolution of the emissions? Do you include a weekly and a diurnal cycle to the emissions?

- **Page 7, line 160-163:** This belongs in the section describing the model setup.

- **Page 7, line 166:** Please also mention briefly how the model performs in simulating the observed meteorology, especially those quantities that would influence the simulation of particles.

- **Page 7, section 3.1.1:** Please revise this section, being clearer about what is observed and how the model results compare to the observations.

- **Page 8, section 3.1.2:** Please choose a section title that fits the contents or adapt the contents to the section title. You do not only speak about PM2.5 and EC, but also include AOD and O3. Please also be more specific, e.g. making clear that those results hold for the three days you simulated.

- **Page 10, line 235:** There can be all kinds of reasons for a reasonable performance in simulating O3, PM2.5 and EC, but it does not automatically lead to the conclusion that the meteorological fields are simulated well. You could be clearer on this if you included some results on evaluating meteorology.

- **Page 10, section 3.2:** Is the beginning of this section more suitable for the introduction? A table summarizing previous results would help here as well.

- **Page 10, section 3.2.1:** not remarkable: how big? Specify this quantitatively.

- **Page 11, line 272:** What is observed?

- **Page 12, line 288:** What is the difference between rural and background sites? How are all sites characterized? Please specify this.

- **Page 12, line 308:** What are the results for Mexico City?

- **Page 13, line 320:** Anthropogenic OA contribution?

- **Page 13, line 324:** Again: this holds for the three days you simulated. How is this representative of springtime/the rest of the year? Is it possible to deduct more general conclusions from the simulation of three days, or can you really only say something about those three days? In order to be able to make recommendations for what would be effective mitigation measures, it would be necessary to simulate at least the whole season.

- **Page 14, line 337:** Verify the OA source - what does this mean?

- **Page 14, line 344:** This is not consistent with what you said before; you did not speak about uncertainties in meteorological fields and emissions. Discuss these uncertainties more in the results section.

- **Page 14, line 353:** This is somewhat confusing. You should mention the factor of 10 explicitly in the results section. You only mention the one of Li et al, which is 7.

- **Page 15, line 364:** ... in the simulated period. Discuss the limitations of simulating a three day period.

- **Page 15, line 370:** is 6% not significant? Reword.

- **Page 15** Outlook: you could be somewhat more specific here on what needs to be done. Also, you might include a small discussion on having a more integrated view on the sources of air pollution (e.g. considering different air pollutants etc) and on assessing mitigation measures, and on what is needed to support the design of mitigation measures.

- **Abstract, line 34/35:** The simulation results will facilitate the design of air pollution control strategies in the basis- I am not sure about this. Either you discuss

this better in the conclusions or you remove it from the abstract/rephrase it considering the points mentioned above.

- **Figure captions 11-13:** Please specify which SOA module was used.

**Technical corrections**

- **line 39:** fine particulate matter
- **line 42:** aerosol concentrations
- **line 45:** components
- **line 59:** air pollution
- **line 62:** plays
- **line 63:** constitutes
- **line 64:** a severe haze pollution event
- **line 65:** contributed
- **line 66:** measurements
- **line 83:** SOA levels have
- **line 98:** Please order the references
- **line 105:** Please add to references and give link in the references section
- **line 126:** grid cells

[Figure]

- **line 194:** it is not reproduced in Figure 5, but the results are shown in Figure 5; please rephrase

- **line 215:** remove well

- **line 275:** since does not fit here, please rephrase this sentence

- **line 324:** Residential emissions are ...

- **line 357:** might be misleading

- **line 357:** devise?

---

## Referee Comment (RC3) · Anonymous Referee #1 · 23 May 2016

General Comments

This manuscript presents the organic aerosol simulations in West China using WRF-Chem. Two types of SOA formation treatments are used, and the simulations are evaluated against monitoring station observations (O3 and PM2.5), campaign sampling site measurements (carbonaceous aerosols), and MODIS AOD retrievals (aerosol optical depth). The authors quantify the SOA fraction in OA and the contribution of different anthropogenic emission sources to OA at different environmental settings (urban, rural and background). The paper stands in a good form with some polishing.

Specific Comments

1. Pull together all measurement data and place in a separate sub-section in Section 2. This may make the structure flow more fluently. The measurement data should

include those used for the model evaluation, such as from the monitoring stations, the field campaign sites, and the MODIS AOD retrievals. Instrumentation for the OA and EC filter field measurements should be described.

2. Add a table summarizing the source-categorized anthropogenic emission in Xi'an and its surroundings in Section 2. This may facilitate the discussions on the OA source apportionment in Section 3.2.4.

3. Clarify Section 3.2. It is not clear to me why certain values of OC/EC and OA/OC ratios are used and how "measured" POA and SOA values are derived in this study. To derive the POA and SOA concentrations from the OA and EC measurements, you need to have the values for the POC/EC, SOC/EC, POA/POC, and SOA/SOC, but in the description there is no presumed value for the SOC/EC ratio. Also please justify the uses of the values of POC/EC (2.4), POA/POC (1.2), and SOA/SOC(1.6), and you may also address that these values may affect the model-measurement comparisons. In addition, it would be interesting to compare the assumed POA/EC and SOA/EC ratios with the simulated counterparts.

4. In the end of Page 15, you rightly point out that "future studies need to be performed to further improve SOA simulations and OA source apportionment". There are many factors contributing to the OA, SOA in particular, simulation uncertainties, including measurements, meteorology, emissions, and SOA formation mechanisms and treatments; even right modeling results might be due to wrong reasons (i.g., right concentrations but wrong O/C ratios). Elaborating a little bit more on what aspects of the SOA modeling can be improved would be insightful.

5. In Section 3.2.4 and Table 2, you may add the OA/PM2.5 fractions, which may provide more scientific information for devising the haze control strategy, since OA is only one important component of the haze in the GZB.

6. P3, lines 65-67. For the upper limit fractions, SOA/PM2.5 has a higher value (77%) than SOA/OA does (71%)? Specify the investigation region in the Sun et al. (2012)

[Figure]

study.

7. Line 85, start a new paragraph from "During the period from 20…".

8. Ls 200-202, the morning elevation could also contributed by the morning rush hour emissions. What do the PM2.5 diurnal profiles look like?

9. Ls 224-232, describe how AOD is estimated in the model.

10. Ls 254-257, conflicting. The difference between 33.6% in T2-SOA and 4.3% in NT-SOA is not "not remarkable".

11. Ls 286 -303, Discussions on the SOA/OA fractions in the two paragraphs overlap, and can be merge them.

12. Ls 331-333, the emission source is one reason; another one can be due to the rapid transport and transformation processes.

13. Ls 341-342, Model results are compared not only against O3 and PM2.5 from the monitoring stations, but also against the PM2.5 carbonaceous components from the OC and EC field filter measurements, and against the MODIS AOD.

14. Page 24 Figure 1 caption, describe the black "circle" (urban borderline?).

Technical comments

1. Line 175, pollution

2. Line 183, concentrations substantially increase to

3. Line 195, delete well

4. Line 198, the air quality with respect to PM2.5

5. L275-276, change to something like "Since the NT-SOA module significantly improves the POA and SOA simulations, we will use the NT-SOA OA simulations for further comparisons and the OA source apportioning. Figure 11…"

6. 310, verify to estimate

7. Line 324, emissions are

---

## Author Comment (AC1) · 14 Jul 2016

**Reply to Anonymous Referee #1**

We thank the reviewer for the careful reading of our manuscript and helpful comments. We have revised the manuscript following the suggestion, as described below.

**General Comments**

This manuscript presents the organic aerosol simulations in West China using WRF-Chem. Two types of SOA formation treatments are used, and the simulations are evaluated against monitoring station observations ($O_3$ and $PM_{2.5}$), campaign sampling site measurements (carbonaceous aerosols), and MODIS AOD retrievals (aerosol optical depth). The authors quantify the SOA fraction in OA and the contribution of different anthropogenic emission sources to OA at different environmental settings (urban, rural and background). The paper stands in a good form with some polishing.

**Specific Comments**

1) Pull together all measurement data and place in a separate sub-section in Section 2. This may make the structure flow more fluently. The measurement data should include those used for the model evaluation, such as from the monitoring stations, the field campaign sites, and the MODIS AOD retrievals. Instrumentation for the OA and EC filter field measurements should be described.

We have included a sub-section in Section 2 summarizing all the measurement data as follows:

*"2.5  Measurement Data*
   *The measurement data include temperature, relative humidity, and wind observations at the Jinghe meteorological station, hourly near-surface $O_3$ and $PM_{2.5}$ concentrations at ambient monitoring stations in Xi'an and surrounding areas, and daily filter measurements of $PM_{2.5}$, OC, and EC at 29 sites in GZB during the field campaign and at Institute of Earth Environment, Chinese Academy of Sciences (IEECAS) in Xi'an during the springtime from 2009 to 2013. The observation sites are categorized into three types regarding their locations: 18 urban sites, 10 rural sites, and 1 background site as shown in Figure 1b. The background site is located in the Qinling Mountains, which is far away from the residential areas. The daily filter samples are obtained on prefired (900 °C, 3 h) 47-mm Whatman QM-A quartz-fiber filters by mini-volume air samplers (Airmetrics, Eugene, OR) at 5 L min$^{-1}$ flow rates.*
   *The measured $PM_{2.5}$ and OC concentrations averaged over the 29 sites in GZB during the*

*study period are compared with the springtime PM$_{2.5}$ and OC observations at IEECAS site from 2009 to 2013 in Figure 3 along with the standard deviations. The springtime PM$_{2.5}$ concentrations at IEECAS site from 2009 to 2013 remain about 150 μg m$^{-3}$ with small fluctuations, which is close to the PM$_{2.5}$ level during the study period. The springtime OC concentrations at IEECAS site from 2009 to 2013 vary from 14 μg m$^{-3}$ to 22 μg m$^{-3}$. The mean OC concentration during the study period is about 19 μg m$^{-3}$, close to the medium level at IEECAS site from 2009 to 2013. Hence, the PM$_{2.5}$ and OC levels during the study period can well represent the springtime PM$_{2.5}$ and OC pollutions in GZB within recent years.*

*The OC/EC ratio approach has been widely employed to evaluate the OA concentration from the filter measured OC and EC concentrations (Strader, 1999; Cao et al., 2003; 2004). The following method is used to derive the 'measured' POA, SOA, and OA mass concentrations from EC and OC filter measurements:*

*POC = EC × (POC/EC ratio)*
*SOC = OC – POC*
*POA = POC × (POA/POC ratio)*
*SOA = SOC × (SOA/SOC ratio)*
*OA = POA + SOA*

*Cao et al. (2007) have analyzed the OC and EC concentrations in 14 cities over China in 2003 and proposed primary OC/EC ratios for different cities in China during winter and summer (Table 3). Numerous studies have been performed to investigate the POA/POC and SOA/SOC ratios (Aiken et al., 2008; Yu et al., 2009; Yu, 2011), which can be used to obtain OA concentrations from measured EC and OC concentrations. In this study, the POC/EC, POA/POC, SOA/SOC ratios are assumed to be 2.4, 1.2, and 1.6, respectively, based on the previous studies (Cao et al., 2007; Aiken et al., 2008; Yu et al., 2009; Yu, 2011). It is worth to note that, these assumed values might affect the model-measurement comparisons.*

*Using the measured EC and OC concentrations, the SOA/EC ratio is calculated:*

*(SOA/EC ratio) = (SOA/SOC ratio) × (SOC/EC ratio)*
*= (SOA/SOC ratio) × [(OC – POC) / EC]*
*= (SOA/SOC ratio) ×[(OC/EC ratio) – (POC/EC ratio)]*

*And the POA/EC ratio is derived as follows:*

*(POA/EC ratio) = (POA/POC ratio) × (POC/EC ratio)"*

MODIS AOD retrievals have been removed from the manuscript according to the suggestion of Anonymous Reviewer #2.

2) Add a table summarizing the source-categorized anthropogenic emission in Xi'an and its surroundings in Section 2. This may facilitate the discussions on the OA source apportionment in Section 3.2.4.

We have included a table and a short description in the last paragraph of Section 2.3 about the anthropogenic emissions in Xi'an and surrounding areas: "*Table 2 presents the primary organic*

*carbon and volatile organic compounds emissions from anthropogenic sources in Xi'an and surrounding areas in April."*

*Table 2 Primary organic carbon and volatile organic compounds emissions from anthropogenic sources in Xi'an and surrounding areas (the area surrounded by the white line in Figure 1c) in April*

| Anthropogenic Emission Sectors | Primary Organic Carbon (Mg) | Volatile Organic Compounds (Mg) |
|---|---|---|
| Industry | 2486.0 | 7634.5 |
| Power Plant | 0.0 | 81.7 |
| Residential | 4988.8 | 1704.4 |
| Transportation | 318.5 | 736.9 |

3) Clarify Section 3.2. It is not clear to me why certain values of OC/EC and OA/OC ratios are used and how "measured" POA and SOA values are derived in this study. To derive the POA and SOA concentrations from the OA and EC measurements, you need to have the values for the POC/EC, SOC/EC, POA/POC, and SOA/SOC, but in the description there is no presumed value for the SOC/EC ratio. Also please justify the uses of the values of POC/EC (2.4), POA/POC (1.2), and SOA/SOC (1.6), and you may also address that these values may affect the model-measurement comparisons. In addition, it would be interesting to compare the assumed POA/EC and SOA/EC ratios with the simulated counterparts.

We have clarified in Section 2.5:

" *The OC/EC ratio approach has been widely employed to evaluate the OA concentration from the filter measured OC and EC concentrations (Strader, 1999; Cao et al., 2003; 2004). The following method is used to derive the 'measured' POA, SOA, and OA mass concentrations from EC and OC filter measurements:*
*POC = EC × (POC/EC ratio)*
*SOC = OC – POC*
*POA = POC × (POA/POC ratio)*
*SOA = SOC × (SOA/SOC ratio)*
*OA = POA + SOA*
*Cao et al. (2007) have analyzed the OC and EC concentrations in 14 cities over China in 2003 and proposed primary OC/EC ratios for different cities in China during winter and summer (Table 3). Numerous studies have been performed to investigate the POA/POC and SOA/SOC ratios (Aiken et al., 2008; Yu et al., 2009; Yu, 2011), which can be used to obtain OA concentrations from measured EC and OC concentrations. In this study, the POC/EC, POA/POC,*

*SOA/SOC ratios are assumed to be 2.4, 1.2, and 1.6, respectively, based on the previous studies (Cao et al., 2007; Aiken et al., 2008; Yu et al., 2009; Yu, 2011). It is worth to note that these assumed values might affect the model-measurement comparisons.*

*Using the measured EC and OC concentrations, the SOA/EC ratio is calculated:*

*(SOA/EC ratio) = (SOA/SOC ratio) × (SOC/EC ratio)*

*= (SOA/SOC ratio) × [(OC – POC) / EC]*

*= (SOA/SOC ratio) ×[(OC/EC ratio) – (POC/EC ratio)]*

*And the POA/EC ratio is derived as follows:*

*(POA/EC ratio) = (POA/POC ratio) × (POC/EC ratio)"*

We have compared the "measured" POA/EC and SOA/EC ratios with simulations in the second paragraph of Section 3.2.1: "*The simulated POA/EC and SOA/EC ratios using the NT-SOA module are 3.23 and 2.23, close to the 'measured' 2.88 and 2.58, respectively. However, the ratios using the T2-SOA module are 6.54 and 0.22, respectively, dramatically deviated from the 'measurements'.*"

4) In the end of Page 15, you rightly point out that "*future studies need to be performed to further improve SOA simulations and OA source apportionment*". There are many factors contributing to the OA, SOA in particular, simulation uncertainties, including measurements, meteorology, emissions, and SOA formation mechanisms and treatments; even right modeling results might be due to wrong reasons (e.g., right concentrations but wrong O/C ratios). Elaborating a little bit more on what aspects of the SOA modeling can be improved would be insightful.

We have elaborated the improvements on SOA modeling in further studies in the last paragraph of Section 4: "*It is worth to note that many factors contribute to the OA and SOA simulation uncertainties, such as measurements, meteorology, emissions, SOA formation mechanisms and treatments et al.; even right modeling results might be caused by wrong reasons (e.g., right concentrations but wrong O/C ratios). To better simulate the SOA formation, the SOA mechanisms and treatments need further revising and improving to reasonably represent OA formation and development in the atmosphere, such as including the oxidation degree, rather than only nine surrogates categorized by saturation concentrations. Additionally, reducing uncertainties from meteorological fields simulations and emissions are also imperative to improve the SOA simulation. In addition to OA, other aerosol species, such as sulfate, nitrate, ammonium, and mineral dust, also play important roles in the haze formation. Further source appointment of those aerosol species is needed to support the design of mitigation strategies.*"

5) In Section 3.2.4 and Table 2, you may add the OA/PM$_{2.5}$ fractions, which may provide more

scientific information for devising the haze control strategy, since OA is only one important component of the haze in the GZB.

We have included the OA/PM$_{2.5}$ fractions in Table 5 and added the following sentence in the second paragraph of Section 3.2.3: "*Table 5 presents OA concentrations and contributions and OA contributions to PM$_{2.5}$ from anthropogenic emissions averaged over the simulation period at urban, rural, and background sites.*"

6) P3, lines 65-67. For the upper limit fractions, SOA/PM$_{2.5}$ has a higher value (77%) than SOA/OA does (71%)? Specify the investigation region in the Sun et al. (2012) study.

We have corrected the sentence in Section 1: "*Huang et al. (2014) have reported that OA constitutes a major fraction (30~50%) of the total PM$_{2.5}$ in Beijing, Shanghai, Guangzhou, and Xi'an during severe haze pollution events (Table 1), and SOA contributed 44~71% of OA mass concentrations.*"

We have specified the investigation region in the Sun et al. (2012) in Section 1: "*Using the ACSM (Aerosol Chemical Speciation Monitor) measurements analyzed by PMF (positive matrix factorization), Sun et al. (2012) have showed that the oxygenated organic aerosols (OOA, a surrogate of SOA) dominate OA composition in Beijing, with a contribution of 64% on average from 26 June to 28 August 2011.*"

7) Line 85, start a new paragraph from "*During the period from 20. . .*".

We have started a new paragraph from "*During the period from 20. . .*" in the last but one paragraph of Section 1.

8) Ls 200-202, the morning elevation could also be contributed by the morning rush hour emissions. What do the PM$_{2.5}$ diurnal profiles look like?

We have revised that sentence in the last paragraph of Section 3.1.1 as "*The PM$_{2.5}$ mass concentrations are generally elevated in the morning during the episode, probably contributed by the weak or calm horizontal winds, slow development of PBL, and the morning rush hour emissions. The PM$_{2.5}$ levels fall down in the afternoon, caused by the enhanced dispersion in the horizontal and vertical directions.*"

9) Ls 224-232, describe how AOD is estimated in the model.

The AOD estimation follows the method in Li et al. (ACP, 2011), in which the extinction efficiency, SSA, and asymmetry factor are calculated using the Mie theory at a given wavelength.

The aerosol spectrum is first divided into 48 bins from 0.002 μm to 2.5 μm, with radius $r_i$. When the bin's radius is less than 0.1 μm, the interval of bins ranges from 0.001 to 0.005 μm. When the bin's radius is greater than 0.1 μm, the interval is increased to 0.025 to 0.25 μm. The aerosols are classified into four types: (1) internally mixed sulfate, nitrate, ammonium, hydrophilic organics, hydrophilic black carbon, and water; (2) hydrophobic organics; (3) hydrophobic black carbon; and (4) other unidentified aerosols. These four kinds of aerosols are assumed to be mixed externally. The aerosol optical thickness (AOT or $\tau_a$) at a given wavelength in a given atmospheric layer $k$ is determined by the summation over all types of aerosols and all bins:

$$\tau_a(\lambda, k) = \sum_{i=1}^{48} \sum_{j=1}^{4} Q_e(\lambda, r_i, j, k) \pi r_i^2 n(r_i, j, k) \Delta z_k$$

where $n(r_i, j, k)$ is the number concentration of j-th kind of aerosols in i-th bin. $\Delta z_k$ is the depth of an atmospheric layer.

Following the suggestion from Anonymous Reviewer #2, we have excluded the AOD comparison in this study.

10) Ls 254-257, conflicting. The difference between 33.6% in T2-SOA and 4.3% in NT-SOA is not "not remarkable".

We have revised that sentence in the first paragraph of Section 3.2.1 as "*The T2-SOA and NT-SOA modules exhibit differently in simulating OA in GZB, as shown in Figure 10a.*"

11) Ls 286-303, Discussions on the SOA/OA fractions in the two paragraphs overlap, and you can merge them.

The paragraphs discuss the SOA fractions (Figure 13) and SOA pathways (Figure 14), respectively. We have moved the following sentence to Section 2.5: "*The observation sites are categorized into three types regarding their locations: 18 urban sites, 10 rural sites, and 1 background site as shown in Figure 1b. The background site is located in the Qinling Mountains, which is far away from the residential areas.*"

12) Ls 331-333, the emission source is one reason; another one can be due to the rapid transport and transformation processes.

We have updated that sentence in the last paragraph of Section 3.2.3: "*It is worth to note that, the OA contributions from residential, transportation, and industry emissions are comparable at the*

*urban and rural sites, which probably is due to the rapid transport and transformation process between urban and rural areas. In addition, the urbanization and industrialization in GZB may also rapidly diminish the OA source difference between the rural and urban areas.*"

13) Ls 341-342, Model results are compared not only against $O_3$ and $PM_{2.5}$ from the monitoring stations, but also against the $PM_{2.5}$ carbonaceous components from the OC and EC field filter measurements, and against the MODIS AOD.

We have clarified in Section 4: "*Model results are compared with the meteorological observations, hourly $O_3$ and $PM_{2.5}$ measurements in Xi'an and surrounding areas, and the $PM_{2.5}$ carbonaceous components from the OC and EC field filter measurements in GZB.*"

14) Page 24 Figure 1 caption, describe the black "circle" (urban borderline?).

We have added in Figure 1 caption: "*The black circle in (b) and (c) shows the ring expressway of Xi'an.*"

**Technical comments**

1) Line 175, pollution

We have changed "pollutions" to "pollution" in the first paragraph of Section 3.1.1.

2) Line 183, concentrations substantially increase to

We have changed sentence in the first paragraph of Section 3.1.1: "*The observed and simulated $O_3$ mass concentrations substantially increase to more than 80 μg m$^{-3}$ at 15:00 BJT with the enhancement of photochemical activities.*"

3) Line 195, delete well

We have deleted "well" in that sentence in the second paragraph of Section 3.1.2.

4) Line 198, the air quality with respect to $PM_{2.5}$

We have changed "*the air quality in Xi'an ...*" to "*the air quality with respect to $PM_{2.5}$ in Xi'an ...*" in the second paragraph of Section 3.1.1.

5) L275-276, change to something like "Since the NT-SOA module significantly improves the POA and SOA simulations, we will use the NT-SOA OA simulations for further comparisons

and the OA source apportioning. Figure 11. . .”

We have changed the sentence in the first paragraph of Section 3.2.2 as "*The NT-SOA module significantly improves POA and SOA simulations, so we use the NT-SOA OA simulations for further comparisons and OA source apportionment. Figure 12 displays the spatial distributions of OA, POA, and SOA simulated by the NT-SOA module against the measurement in GZB.*"

6) 310, verify to estimate

We have changed "verify" to "estimate" in the first paragraph of Section 3.2.3.

7) Line 324, emissions are

We have changed "emission is" to "emissions are" in the second paragraph of Section 3.2.3.

---

## Author Comment (AC2) · 14 Jul 2016

**Reply to Anonymous Referee #2**

We thank the reviewer for the careful reading of our manuscript and helpful comments. We have revised the manuscript following the suggestion, as described below.

This manuscript is clearly written and on an important topic on SOA simulation. It is easy to follow and some of its conclusions are interesting. But I am not happy respecting to following points.

1) The model and the method used in this application are nearly the same as in Li et al. (2011b), and one major conclusion is obvious that NT-SOA produces higher SOA than T2-SOA, because the total amount of material (POA + SVOC + IVOC) introduced in the NT-SOA module is 7.5 times the particle-phase POA emissions. Besides, many studies have already shown that VBS produces higher SOA than the traditional 2-product SOA module.

We have clarified in Section 2: "*The WRF-CHEM model and the SOA modules in the present study are nearly the same as those in Li et al. (2011b), which are briefly introduced in Sections 2.1 and 2.2 for convenience.*"

We have included the following discussions in Section 3.2.1: "*The NT-SOA module produces higher SOA than the T2-SOA module because the total amount of material (POA + Semivolatile-VOC + Intermediate-VOC) included in the NT-SOA module is 7.5 times the particle-phase POA emissions, which is consistent with the conclusion obtained by Li et al. (2011). Besides, many studies have already shown that the VBS approach produces higher SOA than the traditional 2-product SOA module (e.g., Hodzic et al., 2009; Tsimpidi et al., 2010).*"

2) It does not clearly show whether diurnal (time) variation in SOA concentration is improved by including VBS than the traditional 2-product SOA module. NT-SOA module contains more precursors and production processes, is it good or not if "*The diurnal variations from the two models agree well with each other, with peak occurrence during noontime, caused by the enhanced photo chemical activities.*" as shown in Figure 10?

We have rephrased the sentence in the last paragraph of Section 3.2.1: "*Both the two modules produce peak SOA concentrations around noontime, but, apparently, the NT-SOA module yields much more SOA than the T2-SOA module because the NT-SOA module contains more precursors and production processes.*"

3) The model performances are evaluated against surface observed $O_3$, $PM_{2.5}$, EC and OA, but

the importance is not clearly seen to compare with satellite derived AOD. Contribution of OA or SOA to AOD is not significant, and it is not clearly stated in the manuscript how the model calculates AOD, and AOD values depend on aerosol number and size distribution, mixing state and air humidity, which are beyond the scope of this study. So good agreement between simulated and satellite retrieved AOD does not imply the model simulates $PM_{2.5}$ and OA well.

The AOD estimation follows the method in Li et al. (2011, doi:10.5194/acp-11-5169-2011), in which the extinction efficiency, SSA, and asymmetry factor are calculated using the Mie theory at a given wavelength. The aerosol spectrum is first divided into 48 bins from 0.002 μm to 2.5 μm, with radius $r_i$. When the bin's radius is less than 0.1 μm, the interval of bins ranges from 0.001 to 0.005 μm. When the bin's radius is greater than 0.1 μm, the interval is increased to 0.025 to 0.25 μm. The aerosols are classified into four types: (1) internally mixed sulfate, nitrate, ammonium, hydrophilic organics, hydrophilic black carbon, and water; (2) hydrophobic organics; (3) hydrophobic black carbon; and (4) other unidentified aerosols. These four kinds of aerosols are assumed to be mixed externally. The aerosol optical thickness (AOT or $\tau_a$) at a given wavelength in a given atmospheric layer $k$ is determined by the summation over all types of aerosols and all bins:

$$\tau_a(\lambda, k) = \sum_{i=1}^{48} \sum_{j=1}^{4} Q_e(\lambda, r_i, j, k) \pi r_i^2 n(r_i, j, k) \Delta z_k$$

where $n(r_i, j, k)$ is the number concentration of j-th kind of aerosols in i-th bin. $\Delta z_k$ is the depth of an atmospheric layer.

Following the suggestion, we have excluded the AOD comparison part in this study.

---

## Author Comment (AC3) · 14 Jul 2016

**Reply to Anonymous Referee #3**

We thank the reviewer for the careful reading of our manuscript and helpful comments. We have revised the manuscript following the suggestion, as described below.

**General comments**

The paper is well structured and the chosen figures illustrate the results well. The description of the methods used is clear. Previous work is clearly described. The main limitation is that the present study presents simulations of 3 days length, which does not allow for drawing general conclusions, e.g. for a whole season. I suggest that the authors clearly describe this limitation and make it clear throughout the manuscript, and justify why and how the results are still important. Furthermore, I would suggest including a small section on how the model performs in simulating meteorological variables that are important for simulating aerosols. This would help be more consistent on conclusions regarding the simulated meteorology. I suggest to revise the conclusions especially with regards to these two points, aiming at consistency with the results stated in the paper. Finally, I suggest to carefully check the language especially in the introduction of the paper. For further specific comments please see below.

We have included a figure (Figure 3) to compare the measured $PM_{2.5}$ and OC concentrations during the study period with the observed springtime $PM_{2.5}$ and OC concentrations from 2009 to 2013 in Xi'an. We have also added a paragraph in Section 2.5 to discuss the representation of the three-day simulation for the spring: "*The measured $PM_{2.5}$ and OC concentrations averaged over the 29 sites in GZB during the study period are compared with the springtime $PM_{2.5}$ and OC observations at IEECAS site from 2009 to 2013 in Figure 3 along with the standard deviations. The springtime $PM_{2.5}$ concentrations at IEECAS site from 2009 to 2013 remain about 150 μg m$^{-3}$ with small fluctuations, which is close to the $PM_{2.5}$ level during the study period. The springtime OC concentrations at IEECAS site from 2009 to 2013 vary from 14 μg m$^{-3}$ to 22 μg m$^{-3}$. The mean OC concentration during the study period is about 19 μg m$^{-3}$, close to the medium level at IEECAS site from 2009 to 2013. Hence, the $PM_{2.5}$ and OC levels during the study period can well represent the springtime $PM_{2.5}$ and OC pollutions in GZB within recent years.*"

We have also described the limitation of the short simulation period throughout the manuscript and justified the importance of the results in the last paragraph of Section 4: "*It should be noted that, this simulation is conducted during 3 days in spring 2013, so it might be only partially representative of the springtime $PM_{2.5}$ and OC pollutions. Nevertheless, considering that the*

*PM$_{2.5}$ and OC concentrations during the study period are similar to those in the past five years, the model result is still important and provides a reference for springtime OA formation.*"

We have included a figure (Figure 4) and a paragraph to compare the observed and modeled temperature, relative humidity, wind speed and direction in Section 3.1.0: "*Figure 4 shows comparisons of the simulated and observed near-surface temperature, relative humidity, wind speed and direction at the Jinghe meteorological station which is close to Xi'an (the yellow spot in Figure 1c) from 23 to 25 April 2013. The simulated diurnal variations of temperature and relative humidity are in good agreement with the observations. The model also generally well reproduces the wind field compared with observations, except overestimation of the wind speed during the daytime of April 23. No precipitation during the simulation period is observed or modeled.*"

We have also revised the conclusions regarding to the simulation of meteorological variables and the limitation of the 3-day simulation in the first paragraph of Section 4: "*Model results are compared with the meteorological observations, hourly O$_3$ and PM$_{2.5}$ measurements in Xi'an and surrounding areas, and the PM$_{2.5}$ carbonaceous components from the OC and EC field filter measurements in GZB.*" and in the last paragraph of Section 4 (please refer to the above discussion).

We have carefully checked the language in the paper, especially in the introduction. The updated words, phrases and sentences are highlighted.

**Specific comments**

1) Page 2, Line 49: remove complicated

We have removed "complicated" in the second paragraph of Section 1.

2) Page 3: the description of the different contributions of OA to PM$_{2.5}$ is hard to follow, formulate it more understandably; a table summarizing previous results might help.

We have included a table summarizing previous results about different OA contributions to PM$_{2.5}$ in Section 1.

*Table 1 Previous results about different OA contributions*

| Reference | Time Period | City | Description |
|---|---|---|---|
| Huang et al. (2014) | 5–25 January 2013 | Beijing | $OA/PM_{2.5}$ (40.7%) |
| | | Shanghai | $OA/PM_{2.5}$ (48.0%) |
| | | Guangzhou | $OA/PM_{2.5}$ (33.1%) |
| | | Xi'an | $OA/PM_{2.5}$ (30.5%) |
| Sun et al. (2012) | 26 June to 28 August 2011 | Beijing | OOA/OA (64%) |
| Sun et al. (2013) | Summer 2012 | Beijing | (OA in $NR\text{-}PM_1$) / $NR\text{-}PM_1$ (52%) |
| Cao et al. (2013) | MIRAGE-Shanghai 2009 | Shanghai | OC/TC (31%) |

3) Page 3, Line 86: why does the simulation not cover the whole period of the measurement campaign?

Although one-week field campaign was conducted from April 20 to 26, 2013, unfortunately, the occurrence of precipitation in GZB on the first three days caused washout of $PM_{2.5}$ in the atmosphere. Therefore, only the latter period of the campaign is covered in the present study. We have clarified in the last but one paragraph of Section 1: "*due to occurrence of precipitation on the first three days, an episode during 23-25 April was simulated to identify the OA sources in this study*."

4) Page 4, Line 89: OA sources and SOA formation

We have changed "OA and particularly SOA sources and formation" to "OA sources and SOA formation" in the last but one paragraph of Section 1.

5) Page 4, Line 91: Specify the research question, investigate is very broad. Also, OA and SOA are not investigated during springtime, but for three days in springtime. Can you deduct general conclusions from a three-day simulation for springtime? If so, why? Is the simulated period particularly typical for springtime conditions?

We have changed that sentence in the last paragraph of Section 1 as: "*The objective of the present study is to examine the formation and source apportionments of OA and SOA in GZB during three days in the spring of 2013 using the WRF-CHEM model.*"

We have discussed the representation of the three-day simulation for spring. Please refer to the reply to General comments.

6) Page 4, Line 102 and following: You simulate wet deposition, but you do not present any

results on how well precipitation is simulated by the model; or if there even is precipitation. This should be included.

We have clarified in Section 3.1.0: "*No precipitation during the study period is observed or modeled.*"

7) Page 5, Line 134: The jump from the boundary condition resolution to the model resolution is big, and I understand that you do not use a nested setup. How far is the studied area away from the boundaries of the domain? Please specify this in the text.

We have clarified in Section 2.3: "*The distance of GZB from the boundaries of the domain is about 150-200 km (50-70 grid cells).*"

8) Page 6, Line 140: What is the temporal resolution of the emissions? Do you include a weekly and a diurnal cycle to the emissions?

We have clarified in Section 2.3: "*The temporal resolution of the emissions is one-hour, and a weekly and a diurnal cycle are included in the emissions.*"

9) Page 7, Line 160-163: This belongs in the section describing the model setup.

We have moved the sentence to the first paragraph of Section 2.3: "*In the present study, the NCEP ADP Global Surface Observational Weather Data (http://rda.ucar.edu/) in GZB are assimilated in the WRF-CHEM model simulations using the four-dimension data assimilation (FDDA) to improve the simulation of meteorological fields.*"

10) Page 7, Line 166: Please also mention briefly how the model performs in simulating the observed meteorology, especially those quantities that would influence the simulation of particles.

We have included a section to describe the model performance in simulating the observed meteorology, including near-surface temperature, relative humidity, wind speed and direction in Section 3.1.0. Please refer to the reply to General Comment.

11) Page 7, section 3.1.1: Please revise this section, being clearer about what is observed and how the model results compare to the observations.

We have revised Section 3.1.1 to make the comparison between the model results and observations clear: "F*igures 5 and 6 provide the spatial patterns of observed and simulated near-surface $O_3$ and $PM_{2.5}$ mass concentrations at 08:00 and 15:00 Beijing Time (BJT) from April 23 to 25, 2013 in Xi'an and surrounding areas, along with simulated wind fields. The*

*calculated $O_3$ and $PM_{2.5}$ spatial distributions are generally consistent with the observations at the monitoring sites. At 08:00 BJT, the weak near-surface winds and the low planetary boundary layer (PBL, not shown) facilitate the accumulation of pollutants, causing observed and simulated high near-surface $PM_{2.5}$ mass concentrations. The observed and calculated $PM_{2.5}$ mass concentration frequently exceeds 150 μg m$^{-3}$, causing heavy air pollution in Xi'an and surrounding areas. Weak solar insolation slows the photochemical activities and the low PBL is also favorable for buildup of emitted $NO_x$, significantly lowering the $O_3$ level at 08:00 BJT, and both the calculated and observed near-surface $O_3$ concentrations range from 20 to 30 μg m$^{-3}$. At 15:00 BJT, with the development of PBL and enhancement of horizontal winds, the $PM_{2.5}$ mass concentrations are decreased but still remain high level in Xi'an and surrounding areas on April 23 and 24. The simulated strong divergence at 15:00 BJT on April 25 efficiently disperses the $PM_{2.5}$ accumulated in the morning and remarkably improves the air quality in Xi'an and surrounding areas, which is also shown by the observed $PM_{2.5}$ concentrations. The observed and simulated $O_3$ mass concentrations substantially increase to more than 80 μg m$^{-3}$ at 15:00 BJT with the enhancement of photochemical activities."*

12) Page 8, section 3.1.2: Please choose a section title that fits the contents or adapt the contents to the section title. You do not only speak about $PM_{2.5}$ and EC, but also include AOD and $O_3$. Please also be more specific, e.g. making clear that those results hold for the three days you simulated.

We have changed Section 3.1.2 to "Daily $PM_{2.5}$ and EC Simulations in GZB" and the AOD section has been removed following the suggestion from Anonymous Reviewer #2. We have also made clear that the model results only hold for the three days throughout the manuscript.

13) Page 10, Line 235: There can be all kinds of reasons for a reasonable performance in simulating $O_3$, $PM_{2.5}$ and EC, but it does not automatically lead to the conclusion that the meteorological fields are simulated well. You could be clearer on this if you included some results on evaluating meteorology.

We have included a section to describe the model performance in simulating the observed meteorology in Section 3.1.0. Please refer to the reply to General Comment.

14) Page 10, section 3.2: Is the beginning of this section more suitable for the introduction? A table summarizing previous results would help here as well.

We have moved part of the section into the last but one paragraph in Section 1: "*In general, the OC/EC ratio approach can be used to evaluate the OA, POA, and SOA concentrations using the*

*filter measured OC and EC (Strader, 1999; Cao et al., 2003; 2004)."* and the rest of the section into Section 2.5:

" *The OC/EC ratio approach has been widely employed to evaluate the OA concentration from the filter measured OC and EC concentrations (Strader, 1999; Cao et al., 2003; 2004). The following method is used to derive the "measured" POA, SOA, and OA mass concentrations from EC and OC filter measurements:*
    *POC = EC × (POC/EC ratio)*
    *SOC = OC – POC*
    *POA = POC × (POA/POC ratio)*
    *SOA = SOC × (SOA/SOC ratio)*
    *OA = POA + SOA*
*Cao et al. (2007) have analyzed the OC and EC concentrations in 14 cities over China in 2003 and proposed primary OC/EC ratios for cities in China during winter and summer (Table 3). Numerous studies have been performed to investigate the POA/POC and SOA/SOC ratios (Aiken et al., 2008; Yu et al., 2009; Yu, 2011), which can be used to obtain OA concentrations from measured EC and OC concentrations. In this study, the POC/EC, POA/POC, SOA/SOC ratios are assumed to be 2.4, 1.2, and 1.6, respectively, based on the previous studies (Cao et al., 2007; Aiken et al., 2008; Yu et al., 2009; Yu, 2011). It is worth to note that, these assumed values might affect the model-measurement comparisons.*
    *Using the measured EC and OC concentrations, the SOA/EC ratio is calculated:*
    *(SOA/EC ratio) = (SOA/SOC ratio) × (SOC/EC ratio)*
                      *= (SOA/SOC ratio) × [(OC – POC) / EC]*
                      *= (SOA/SOC ratio) ×[(OC/EC ratio) – (POC/EC ratio)]*
*And the POA/EC ratio is derived as follows:*
    *(POA/EC ratio) = (POA/POC ratio) × (POC/EC ratio)*"

We have included a table summarizing previous results about primary OC/EC ratios over China in 2003 proposed by Cao et al. (2007) in Table 3.

*Table 3 The primary OC/EC ratios over China in 2003 proposed by Cao et al. (2007)*

|  | Northern Cities | | Southern Cities | |
| --- | --- | --- | --- | --- |
|  | *Winter* | *Summer* | *Winter* | *Summer* |
| *POC/EC ratio* | *2.81* | *1.99* | *2.10* | *1.29* |

15) Page 10, section 3.2.1: not remarkable: how big? Specify this quantitatively.

We have clarified in the first paragraph of Section 3.2.1: "*The T2-SOA and NT-SOA modules exhibit differently in simulating OA in GZB, as shown in Figure 10a. The T2-SOA module*

17

*overestimates the observed OA concentrations by about 33.6% averaged over the 29 sites, and the NT-SOA modules underestimates the observation by about 4.3%.* ”

16) Page 11, Line 272: What is observed?

We have added the observation in the last paragraph of Section 3.2.1: “*The NT-SOA module remarkably improves the SOA yields during the entire episode to around 7.2 μg $m^{-3}$ compared with the observed SOA of 8.2 μg $m^{-3}$ averaged over the 29 sites, about tenfold increase compared with the simulations from the T2-SOA module.*”

17) Page 12, Line 288: What is the difference between rural and background sites? How are all sites characterized? Please specify this.

We have specified this in the first paragraph of Section 2.5: “*The observation sites are categorized into three types regarding their locations: 18 urban sites, 10 rural sites, and 1 background site as shown in Figure 1b. The background site is located in the Qinling Mountains, which is far away from the residential areas.*”

18) Page 12, Line 308: What are the results for Mexico City?

We have updated that sentence in the last paragraph of Section 3.2.2 as “*The GSOA constitutes about 10% of SOA mass in the afternoon at urban sites, close to the 9.6% contribution of the observed SOA mass in urban area of Mexico City (Li et al., 2011b).*”

19) Page 13, Line 320: Anthropogenic OA contribution?

We have updated the sentence in the second paragraph of Section 3.2.3: “*During the simulation episode, anthropogenic emissions play a predominant role in the OA formation at the urban and rural sites, with the OA contribution of 82.4% (18.2 μg $m^{-3}$) and 77.3% (12.8 μg $m^{-3}$), respectively.*”

20) Page 13, Line 324: Again: this holds for the three days you simulated. How is this representative of springtime/the rest of the year? Is it possible to deduct more general conclusions from the simulation of three days, or can you really only say something about those three days? In order to be able to make recommendations for what would be effective mitigation measures, it would be necessary to simulate at least the whole season.

We have emphasized that the result is for the three days we have simulated in the second paragraph of Section 3.2.3: “*During the simulation episode, anthropogenic emissions play a predominant role in the OA formation at the urban and rural sites, with the OA contribution of*

*82.4% (18.2 μg m⁻³) and 77.3% (12.8 μg m⁻³), respectively. At the background site, the OA contribution from anthropogenic emissions is close to 60% (4.7 μg m⁻³), showing that the background area is still substantially influenced by human activities despite the far distance from the urban area. Residential emissions are the most important anthropogenic OA source at the urban and rural sites in the present study, with the OA contribution close to 50%, indicating that reducing residential emission is an efficient approach for OA mitigation in GZB.*" We have also discussed the limitation of the short simulation period and the importance of the result in this study in the last paragraph of Section 4. Please refer to the reply to General Comment.

21) Page 14, Line 337: Verify the OA source - what does this mean?

We have revised that sentence in the first paragraph of Section 4: "*A 3-day episode from 23 to 25 April 2013 is simulated in the Guanzhong basin, China using the WRF-CHEM model to examine OA sources and SOA formation.*"

22) Page 14, Line 344: This is not consistent with what you said before; you did not speak about uncertainties in meteorological fields and emissions. Discuss these uncertainties more in the results section.

We have included the meteorological field comparison between the model and the observation in Section 3.1.0 (please refer to the reply to General Comments) and further clarified in Section 3.1.1: "*The PM₂.₅ mass concentrations are generally elevated in the morning during the episode, probably contributed by the weak or calm horizontal winds, slow development of PBL, and the morning rush hour emissions. The PM₂.₅ levels fall down in the afternoon, caused by the enhanced dispersion in the horizontal and vertical directions. The deviations between the model results and observations might be caused by the rapid changes in anthropogenic emissions that are not reflected in the current emission inventories, or the uncertainties in the meteorological filed simulations (Bei et al., 2008; 2010; 2012).*"

23) Page 14, Line 353: This is somewhat confusing. You should mention the factor of 10 explicitly in the results section. You only mention the one of Li et al, which is 7.

We have rephrased the sentence in the first paragraph of Section 3.2.1: "*The T2-SOA module overestimates the measured POA by around 132.0%, and only explains about 9.4% of the observed SOA concentration; the SOA underestimation of the T2-SOA module reaches a factor of 10.*"

24) Page 15, Line 364: ... in the simulated period. Discuss the limitations of simulating a three-day period.

We have clarified in the last but one paragraph of Section 4: "*During the simulated period, the oxidation of anthropogenic and biogenic VOCs and the irreversible uptake of dicarbonyl compounds do not constitute an important SOA formation pathway in GZB, with the SOA contributions less than 10% generally.*" and discussed the limitations of simulating a three-day period in the last paragraph of Section 4: "*It should be noted that, this simulation is conducted during 3 days in spring 2013, so it might be only partially representative of the springtime PM$_{2.5}$ and OC pollutions. Nevertheless, considering that the PM$_{2.5}$ and OC concentrations during the study period are similar to those in the past five years, the model result is still important and provides a reference for springtime OA formation.*"

25) Page 15, Line 370: is 6% not significant? Reword.

We have rephrased the sentence as "*The OA contribution from industry emissions is less than 6.1% in GZB*" in the last but one paragraph of Section 4.

26) Page 15 Outlook: you could be somewhat more specific here on what needs to be done. Also, you might include a small discussion on having a more integrated view on the sources of air pollution (e.g. considering different air pollutants etc.) and on assessing mitigation measures, and on what is needed to support the design of mitigation measures.

We have included discussions in the last paragraph of Section 4: "*It is worth to note that many factors contribute to the OA and SOA simulation uncertainties, such as measurements, meteorology, emissions, SOA formation mechanisms and treatments et al.; even right modeling results might be caused by wrong reasons (e.g., right concentrations but wrong O/C ratios). To better simulate the SOA formation, the SOA mechanisms and treatments need further revising and improving to reasonably represent OA formation and development in the atmosphere, such as including the oxidation degree, rather than only nine surrogates categorized by saturation concentrations. Additionally, reducing uncertainties from meteorological fields simulations and emissions are also imperative to improve the SOA simulation. In addition to OA, other aerosol species, such as sulfate, nitrate, ammonium, and mineral dust, also play important roles in the haze formation. Further source appointment of those aerosol species is needed to support the design of mitigation strategies.*"

27) Abstract, Line 34/35: The simulation results will facilitate the design of air pollution control strategies in the basis- I am not sure about this. Either you discuss this better in the conclusions or you remove it from the abstract/rephrase it considering the points mentioned above.

We have removed the sentence in the Abstract.

28) Figure captions 11-13: Please specify which SOA module was used.

We have specified the SOA module used in the figure captions for Figures 12-14 (original Figures 11-13):

*Figure 12 Spatial distributions of the NT-SOA module calculated (contours) and filter observed (squares) daily OA (left column), POA (middle column), and SOA (right column) mass concentrations from 23 to 25 April 2013. Red squares in (g), (h), and (i) show the location of the background site.*

*Figure 13 Comparisons between the NT-SOA module predicted and filter measured daily SOA mass fraction in OA at urban, rural and background sites during the simulation episode.*

*Figure 14 NT-SOA module simulated contributions of different formation pathways to SOA levels averaged over the simulation episode at (a) urban, (b) rural, and (c) background sites.*

**Technical corrections**

1) Line 39: fine particulate matter

We have changed "fine particulate matters" to "fine particulate matter" in the first paragraph of Section 1.

2) Line 42: aerosol concentrations

We have changed "aerosols" to "aerosol concentrations" in the first paragraph of Section 1.

3) Line 45: components

We have changed "component" to "components" in the second paragraph of Section 1.

4) Line 59: air pollution

We have changed "air pollutions" to "air pollution" in the third paragraph of Section 1.

5) Line 62: plays

We have changed "play" to "plays" in the third paragraph of Section 1.

6) Line 63: constitutes

We have changed "constitute" to "constitutes" in the third paragraph of Section 1.

7) Line 64: a severe haze pollution event

We have changed "during the severe haze pollution event" to "during a severe haze pollution event" in the third paragraph of Section 1.

8) Line 65: contributed

We have changed "contribute" to "contributed" in the third paragraph of Section 1.

9) Line 66: measurements

We have changed "measurement" to "measurements" in the third paragraph of Section 1.

10) Line 83: SOA levels have

We have changed "High SOA level has" to "High SOA levels have" in the last but two paragraph of Section 1.

11) Line 98: Please order the references

We have reordered the references in the first paragraph of Section 2.1 as "*Li et al. (2010; 2011a; 2011b; 2012)*".

12) Line 105: Please add to References and give link in the references section

We have included the reference in the first paragraph of Section 2.1: "*(Nenes et al., 1998)*" and provided the link in the reference section

*Nenes, A., Pilinis, C., Pandis, S. N.: ISORROPIA: A New Thermodynamic Equilibrium Model for Multiphase Multicomponent Inorganic Aerosols, Aquat. Geochem., 4(1), 123-152, http://nenes.eas.gatech.edu/ISORROPIA/, 1998.*"

13) Line 126: grid cells

We have changed "grids" to "grid cells" in the first paragraph of Section 2.3.

14) Line 194: it is not reproduced in Figure 5, but the results are shown in Figure 5; please rephrase

We have rephrased the sentence in the second paragraph of Section 3.1.1 as: "*Figure 7 shows that the variations of observed PM$_{2.5}$ are reasonably reproduced by the model …*"

15) Line 215: remove well

We have removed "*well*" in the second paragraph of Section 3.1.2.

16) Line 275: since does not fit here, please rephrase this sentence

We have rephrased the sentence in the first paragraph of Section 3.2.2 as: "*The NT-SOA module significantly improves POA and SOA simulations, so we use the NT-SOA OA simulations for further comparisons and the OA source apportioning.*"

17) Line 324: Residential emissions are ...

We have changed "*The residential emission is…*" to "*Residential emissions are…*" in the second paragraph of Section 3.2.3.

18) Line 357: might be misleading

We have changed "might cause misleading" to "*might be misleading*" in the third paragraph of Section 4.

19) Line 357: devise?

We have changed "*devise*" to "*develop*" in the third paragraph of Section 4.